# Evaluating different methods for elevation calibration of MAX-DOAS instruments during the CINDI-2 campaign

Sebastian Donner[1], Jonas Kuhn[1,2], Michel Van Roozendael[3], Alkiviadis Bais[4], Steffen Beirle[1], Tim Bösch[5], Kristof Bognar[6], Ilya Bruchkouski[7], Ka Lok Chan[8], Steffen Dörner[1], Theano Drosoglou[4], Caroline Fayt[3], Udo Frieß[2], François Hendrick[3], Christian Hermans[3], Junli Jin[9], Ang Li[10], Jianzhong Ma[11], Enno Peters[5,*], Gaia Pinardi[3], Andreas Richter[5], Stefan F. Schreier[12], André Seyler[5], Kimberly Strong[6], Jan-Lukas Tirpitz[2], Yang Wang[1], Pinhua Xie[10], Jin Xu[10], Xiaoyi Zhao[6,**], and Thomas Wagner[1]

[1]Max Planck Institute for Chemistry, Mainz, Germany
[2]Institute of Environmental Physics, University of Heidelberg, Heidelberg, Germany
[3]BIRA-IASB - Royal Belgian Institute for Space Aeronomy, Brussels, Belgium
[4]Laboratory of Atmospheric Physics, Aristotle University of Thessaloniki, Thessaloniki, Greece
[5]Institute of Environmental Physics, University of Bremen, Bremen, Germany
[6]Department of Physics, University of Toronto, Toronto, Canada
[7]National Ozone Monitoring Research and Education Center BSU, Minsk, Belarus
[8]Meteorological Insitute, Ludwig-Maximilians-Universität, Munich, Germany
[9]CMA Meteorological Observation Center, Beijing, China
[10]Anhui Institute of Optics and Fine Mechanics, Hefei, China
[11]Chinese Academy of Meteorological Sciences, Beijing, China
[12]Institute of Meteorology, University of Natural Resources and Life Sciences, Vienna, Austria
[*]now at: Institute for the Protection of Maritime Infrastructures, German Aerospace Center (DLR), Bremerhaven, Germany
[**]now at: Air Quality Research Division, Environment and Climate Change Canada, Toronto, Canada

*Correspondence to:* Sebastian Donner (sebastian.donner@mpic.de)

**Abstract.** We present different methods for in-field elevation calibration of MAX-DOAS (Multi AXis Differential Optical Absorption Spectroscopy) instruments that were applied and inter-compared during the second Cabauw Intercomparison campaign for Nitrogen Dioxide measuring Instruments (CINDI-2). One necessary prerequisite of consistent MAX-DOAS retrievals is a precise and accurate calibration of the elevation angles of the different measuring systems. Therefore, different methods for this calibration were applied to several instruments during the campaign and the results were inter-compared.

This work first introduces and explains the different methods, namely far and near lamp measurements, white stripe scans, horizon scans and sun scans, using data and results for only one (mainly the MPIC) instrument. In the second part, the far lamp measurements and the horizon scans are examined for all participating groups. Here, the results for both methods are first inter-compared for the different instruments and secondly, the two methods are compared amongst each other.

All methods turned out to be well-suited for the calibration of the elevation angles of MAX-DOAS systems, with each of them having individual advantages and drawbacks. Considering the results of this study, the systematic uncertainties of the methods can be estimated as ±0.05° for the far lamp measurements and the sun scans, ±0.25° for the horizon scans, and around ±0.1° for the white stripe and near lamp measurements. When comparing the results of far lamp and horizon scan

measurements, a spread of around 0.9° in the elevation calibrations is found between the participating instruments for both methods. This spread is on the order of a typical field of view (FOV) of a MAX-DOAS instrument and therefore, affecting the retrieval results. Further, a consistent (wavelength dependent) offset of 0.32° and 0.40° between far lamp measurements and horizon scans is found, which can be explained by the fact that, despite the flat topography around the measurement site,

obstacles such as trees might mark the visible horizon during daytime. The observed wavelength dependence can be explained by surface albedo effects. Lastly, the results are discussed and recommendations for future campaigns are given.

## 1   Introduction

Multi AXis Differential Optical Absorption Spectroscopy (MAX-DOAS) is a well established method of ground-based remote sensing of trace gases and aerosols. MAX-DOAS instruments measure spectra of scattered sunlight at different (mostly low)

elevation angles. Therefore, they have a high sensitivity to trace gases and aerosols located close to the surface (e.g., Hönninger and Platt, 2002; Hönninger et al., 2004; Irie et al., 2008; Van Roozendael et al., 2004; Wagner et al., 2004, 2011; Wittrock et al., 2004). Such measurements allow the retrieval of vertical profiles of trace gases and aerosol extinction as well as column properties such as vertical column densities (VCDs) and aerosol optical depths (AODs) (e.g., Frieß et al., 2006; Irie et al., 2008; Clémer et al., 2010; Wagner et al., 2011).

For the retrieval of these quantities, it is crucial to accurately know the measurement geometry, namely the solar zenith angle, the relative solar azimuth angle and the telescope's elevation angle (Hönninger et al., 2004). While the solar zenith angle can be computed rather easily, it is more difficult to determine the relative solar azimuth angle and to calibrate the telescope's elevation angles. Although the relative azimuth has at least the same uncertainties as the elevation angle, it has a much weaker effect on the measurements (when assuming horizontal homogeneity of the trace gas distributions). An analysis of the CINDI-2

data set shows that for low elevation angles wrong pointing has a large impact on the retrieved trace gas differential slant columns (dSCDs) which are the basic quantity obtained by MAX-DOAS (Hönninger et al., 2004). Panel (a) of Fig. 1 shows the mean dependence of the dSCDs on the elevation angle for the whole campaign and for different trace gases. Each curve was normalised to the mean dSCD of the respective elevation sequence. Panel (b) shows the relative changes in the retrieved trace gas dSCDs per degree pointing error for the same species. The curves were obtained by calculating the derivative of the

dSCD curves from panel (a) with respect to the elevation angle and dividing the results by the dSCDs at the corresponding elevations. Although this approach remains qualitative, it shows clearly under which conditions pointing errors can lead to substantial biases in the dSCDs. As an example, an error of 1° in the telescope's elevation close to 0° elevation can lead to an error of around 20 % in the retrieved $NO_2$ dSCD, since the sensitivity of MAX-DOAS measurements depends strongly on the elevation angle and $NO_2$ is mainly located close to the surface.

Since instruments and retrieval algorithms have improved significantly in recent years, uncertainties in the elevation calibration now have a larger relative impact on the retrieval results and can become the dominating error source. Therefore, the calibration of the elevation angles of MAX-DOAS instruments has become an important topic for instrument operators (e.g., Roscoe et al., 2010; Piters et al., 2012). In principle, these calibrations can be best done in the laboratory under stable

and controlled conditions, where fixed target points are used as references and the corresponding elevations can be calibrated accordingly. In particular, the field of view (FOV) should be determined already in the laboratory. Nevertheless, elevation calibration in the field is indispensable, because during transport from the laboratory to the field and during installation on the measurement site, it is likely that the instrument characteristics might change. In the past, however, when the instruments were brought to the field, only rarely (if at all) the accuracy of the a-priori elevation angle calibration was checked under real measurement conditions.

In this work, different methods for in-field elevation calibration of MAX-DOAS instruments, which were applied by numerous groups during the CINDI-2 campaign, are presented, evaluated and compared amongst each other. Furthermore, recommendations for the setup of MAX-DOAS measurements are derived based on the results of the comparison. This work focuses on the details of the elevation calibration procedure. An assessment of the instrument performance is given in Kreher et al. (2019).

The paper is structured as follows. First, a short overview of the CINDI-2 campaign and the participating instruments is given in Sect. 2. Section 3 introduces and examines the different methods of elevation calibration for one (mainly the instrument of the Max Planck Institute for Chemistry (MPIC)) instrument and a first comparison between the different methods is given. Comparisons between the participating instruments for the different methods are presented in Sect. 4. Additionally, the methods are compared amongst each other in detail and an assessment of the in-field FOV determination is presented. A final discussion is presented in Sect. 5, together with conclusions and recommendations for upcoming MAX-DOAS measurements.

## 2 Campaign overview and MAX-DOAS instruments

### 2.1 The CINDI-2 campaign

The second Cabauw Intercomparison campaign for Nitrogen Dioxide measuring Instruments (CINDI-2) took place in September 2016 at the Cabauw Experimental Site for Atmospheric Research (CESAR) in the Netherlands. It was a follow up to the CINDI campaign, which took place at the same site in June 2009 (Roscoe et al., 2010; Piters et al., 2012; Pinardi et al., 2013; Frieß et al., 2016). The main objective of the CINDI-2 campaign was the inter-comparison of different ground-based remote sensing - mostly MAX-DOAS - instruments, including several calibration exercises to harmonise the measurements of the different instruments. For the interpretation of the trace gas (e.g. $NO_2$) inter-comparisons, an accurate and consistent elevation calibration is essential. Therefore, an elevation calibration exercise was included in the campaign plan. More detailed descriptions of the CINDI-2 campaign and its objectives are given in Kreher et al. (2019) and Apituley et al. (2019).

Many instruments which were operated during the campaign also participated in different elevation calibration exercises that were conducted throughout CINDI-2 and are summarised in this work. In the following subsection, some technical details of the different instruments are presented.

## 2.2 Participating MAX-DOAS instruments

Several instruments from variuos groups participated in the elevation calibration exercise. Since horizon scans were part of the standardised measurement protocol (Kreher et al., 2019; Apituley et al., 2019), they were performed by in total 28 instruments. However, only 12 instruments from 11 groups participated actively in the other calibration exercise, namely the far lamp measurements. Therefore, only these are included in Table 1 which provides an overview on the key properties of the instruments which are of relevance for this study. Further, this table defines instrument acronyms and names. Table 1 is based on the CINDI-2 planning document and information given in Kreher et al. (2019), where additional details on the instruments are provided. Finally, this table indicates the variety and different properties of the participating instruments.

## 3 Methods of elevation calibration

This section introduces the different methods for the calibration of the elevation angles of MAX-DOAS instruments that were applied by at least one group during the CINDI-2 campaign. First, the common general approach which is underlying all methods is explained (subsection 3.1), followed by detailed descriptions and evaluations of the individual methods (subsections 3.2 to 3.6). Subsection 3.7 gives a consistency check between the different methods using data of the MPIC instrument.

## 3.1 General approach

As already mentioned, MAX-DOAS instruments which are brought to field campaigns typically have an a-priori calibration of the elevation angles which was obtained in the laboratory. Different procedures of laboratory elevation calibrations were used by the groups participating in this study. A large fraction of the groups verified the alignment of the telescope and the optical system through retro-illumination of the quartz fibre bundle and measurement of the position of the resulting light spot on a wall, where target points were located. Others, used high precision water levels to check this alignment or even performed no laboratory elevation calibrations. The groups reported that the accuracies and precisions of this procedures are in a range from roughly 0.1° to 1°. However, all this might be overruled by uncertainties introduced during transport or on the measurement site (tripod movements, building movements, non-perfect horizontal alignment of telescope head on tripod etc.). Therefore, for high quality measurements this a-priori calibration has to be checked and (if needed) to be adjusted in field. If the optical axis would be precisely known, a water level would be completely sufficient to calibrate the elevation angles. But this assumption is not always fulfilled, e.g. because the fibre bundle is not perfectly centered. For those reasons, five different methods for the in-field calibration of the elevation angles of MAX-DOAS systems are described and evaluated in the following sections. All of them use the same basic principle which is described in this section.

First, a specific target is placed in front of the optical unit of a MAX-DOAS instrument (see Fig. 2). The elevation angle $\beta$ of this target relative to the horizontal line of the telescope unit has to be determined very accurately in order to use this elevation as a reference. The horizontal line corresponds to the line of sight of the instrument at 0° elevation. Next, an elevation scan across the target is performed and the apparent elevation angle $\widehat{\alpha}$ of the target is retrieved using the measured intensities at

different elevation angles $\alpha_i$. Here, $\alpha_i$ denotes elevation angles measured relative to the a-priori elevation calibration of the instrument. In that way, an elevation offset $\gamma = \beta - \widehat{\alpha}$ can be calculated and used to correct the elevation angles measured relative to the a-priori elevation calibration of the instrument ($\alpha' = \alpha + \gamma$). Further, this offset $\gamma$ can be used to adjust the a-priori elevation calibration of the MAX-DOAS system by shifting the a-priori horizontal line by $-\gamma$. However, it should be

noted that this principle used by all the different methods described in this work only calibrates one specific elevation angle, usually 0°. Thus, the accuracy of the other elevations depends on the internal accuracy of the motor and its steering unit. Many instruments (e.g. the CMA MiniDOAS instrument and the MPIC Tube MAX-DOAS instrument) approach the other elevation angles by moving the telescope by a defined number of motor steps (per degree) relative to the motor position that corresponds to the horizontal line of the telescope (0° elevation). Others however (e.g. the 2D-EnviMeS instruments), use an inclinometer

inside the telescope unit which also enables these instruments to actively correct their elevation angles for possible deviations. According to the instrument's manufacturer the accuracy of the inclinometers used for the EnviMeS instruments is 0.1° and the precision is 0.03°. Many other methods to deal with this internal elevation calibration are possible, but this topic is not discussed in more detail in this paper. Further, it should be mentioned that the elevation calibration might be different for different azimuthal directions which is important for 2D instruments.

A sketch of the general measurement setup can be found in Fig. 2, further, a graphic definition of all relevant angles is provided in this figure. In the following sections this principle is applied to different target types and the results are evaluated.

## 3.2  Far lamp measurements

### 3.2.1  Approach

For the first method, an artificial light source is used as target that is located at a far distance (around 1 to 2 km) from the

instrument's telescope and typically close to the visible horizon. Since this method uses an artificial light source, the elevation scan across this target has to be done during nighttime.

During the CINDI-2 campaign, a xenon lamp was used as light source and was placed at around 1280 m distance from the measurement site in the main viewing direction of the MAX-DOAS instruments at an azimuth angle of 287° (upper panel of Fig. 3). The lamp was put in the focal point of a large aperture lens with a diameter of 17 cm which was achieved by minimising

the size of the light beam (this was already done prior to the campaign). Then the lamp was manually directed towards the campaign site. Here, it should be noted that the exact pointing is not critical as long as the instruments are located within the light cone. It was assumed that the diameter of the lens is homogeneously bright. However, also this assumption is not a critical point, because the angle under which the full lens is seen from the campaign site is smaller than 0.01°.

Using the connected water channels located next to both the measurement site and the lamp site, we could determine and

mark the vertical position of the lamp at the measurement site (lamp mark in the lower panel of Fig. 3). Therefore, the light of a laser level was projected onto a folding rule which was placed in the nearby channels. In that way, first the height difference between lamp and the channel's water surface could be determined. Since all channels were connected to each other (except one step which was determined in the same way), the lamp position could be marked on the containers as indicated in the

lower panel of Fig. 3. Thus, the height difference $\Delta h$ between the optical units of the instruments and the lamp mark could be determined. This $\Delta h$ was then used to infer the expected elevation angles $\beta$ of the lamp relative to the horizontal lines of the optical units of the individual MAX-DOAS systems. The layout of the measurement conditions and the measurement geometry are summarised in Fig. 3.

Such xenon lamp measurements were done on several days (8[th], 10[th], 13[th] and 19[th] September) throughout the campaign, although not all instruments participated on all nights. In the next section, the analysis of the lamp measurements is explained in more detail using data from the MPIC instrument.

### 3.2.2    Results for the MPIC instrument

Data from the far lamp measurements are available for four nights for the MPIC instrument. On all of these nights, a fixed

elevation calibration (same 0° motor position) was used and the scan resolution was 0.02° (except on 8[th] September, when the scan resolution was 0.1° as indicated in the last column of Table 2). For the MPIC instrument the pre-calibration of the elevations was done using a water level during the setup of the instrument. Then finer adjustments were performed using the results of the far lamp scans from 7[th] (in this night the lamp measurements were tested by our group with a scan resolution of 0.1° but the scanning was done manually), 8[th] and 10[th] September. All elevations of the MPIC instrument in this paper are

given relative to the elevation calibration which was obtained by these finer adjustments and which was finally used for the regular measurements.

     In the following, the analysis is done for three wavelengths, which are distributed over the detector range of the instrument and correspond to strong emission lines of the xenon lamp. An example spectrum of the xenon lamp which was measured on 13[th] September is shown in panel (a) of Fig. 4. The three distinct emission lines at 365.16 nm, 404.90 nm and 435.96 nm that

were used for the analysis are clearly visible.

     As first step of the analysis, the measured intensities are normalised with respect to their total integration time and linearly interpolated between the two detector pixels closest to each of the three selected wavelengths. These intensities are then plotted against the elevation angle for the different scans. As an example, the intensity curve at 435.96 nm obtained for 13[th] September is shown in Fig. 4(b). The curve obviously shows a minimum where a maximum would be expected if we assume a Gaussian-

shaped curve. However, we can understand this feature when we take into consideration that in the focal point of the telescope, a quartz glass fibre bundle is mounted as illustrated in Fig. 5. First, we calculate the size of the image of the xenon lamp inside the instrument's telescope (yellow spot in Fig. 5). Given the geometry of the measurement setup, namely the diameter of the xenon lamp and the dimensions of the telescope, this leads to an image size of around 7 μm at the entrance of the fibre bundle. Taking into account that the glass fibre bundle consists of four individual fibres with a light-conducting diameter of 200 μm

each, the obtained image size is only 3.5 % of a single fibre diameter. In that way, it is possible that the image of the lamp hits the space between the individual fibres when performing an elevation scan (dashed line in Fig. 5 indicates the idealised scan axis) and therefore an intensity minimum is found when exactly pointing at the light source. These calculations were done assuming idealised conditions (fibre exactly located in the focus, no aberration of the lens etc.) and the resulting image of the xenon lamp would lead to a much more pronounced and wider minimum than the one in Fig. 4(b). However, in reality the lens

has an aberration and the fibre bundle might be located not exactly in the focus of the lens, further, the scan axis might not pass through the center of the fibre bundle. These effects lead to a less symmetric intensity distribution which does not reach zero intensity at its center (Fig. 4(b)).

In order to determine the elevation angle $\widehat{\alpha}$ under which the xenon lamp could be seen, the center of the intensity curve (dashed blue line in Fig. 4(b)) is calculated using the center of mass formula:

$$\widehat{\alpha} = \frac{\sum s_i \cdot \alpha_i}{\sum s_i}, \tag{1}$$

where $s_i$ denotes the intensity measured at the elevation angle $\alpha_i$. Using this equation yields a lamp elevation of $0.02°$ for the intensity curve shown in panel (b) of Fig. 4. Here, it should be noted that Fig. 4(b) shows the intensity curve of an elevation scan that was performed by approaching the elevation angles from below. For scans where the angles were approached from above, the centers are found consistently at lower elevations by around $0.4°$. Because of that, we assured that all elevation angles were approached from below for the other calibration exercises since this was the scanning direction prescribed by the regular measurement protocol. It should be mentioned that depending on the kind of stepper/motor not all instruments suffered from such backlash issues. Some, actively corrected for this by using inclinometers (e.g. LMU and IUP-HD) or active sun trackers (e.g. BIRA). Besides that, most of the instruments which experienced backlash issues solved them by simply scanning always from the same direction (in elevation and azimuth direction). The effect of backlash (maximum difference between both scanning directions) ranges from fractions of a degree to roughly $1°$. While this effect is very important for the elevations of the instruments, the effect has a much smaller influence on the measurements in the azimuth direction.

Equation (1) is used to calculate the centers of the intensity curves for all three wavelengths and all four days. The corresponding lamp positions are summarised in Table 2. Taking into account that the minimum motor step size is $0.01°$, the different values are consistent with each other within the span from $-0.01$ to $0.02°$ (excluding $8^{\text{th}}$ September, when the scan resolution was only $0.1°$). Here, it should be noted that the center of a Gaussian fit (see red fit curve in Fig. 4(b)) yields consistent lamp elevations compared to the center of mass approach which was applied here. Therefore, also for the MPIC instrument, a Gaussian fit is used in Sect. 4.1, where the lamp scans of all instruments are analysed in a consistent way.

As already mentioned above, the position of the artificial light source relative to the instrument has to be known very accurately in order to calibrate the elevation angles of the MAX-DOAS system. Based on the setup summarised in Fig. 3, an expected lamp elevation of around $-0.04°$ is obtained, when using an estimated height difference, $\Delta h$, of $1\,\text{m}$ between the xenon lamp and the telescope unit. The total error in the determination of the lamp mark (error of $\pm 0.2\,\text{m}$) and the height difference $\Delta h$ (error of $\pm 0.3\,\text{m}$) is estimated to be around $\pm 0.5\,\text{m}$ which translates to an uncertainty of $\pm 0.02°$ in the expected lamp position. Further, it should be noted that the Earth's curvature at a distance of $1280\,\text{m}$ corresponds already to $-0.011°$ and is therefore not negligible. Adding this offset to the obtained lamp elevation, the MPIC MAX-DOAS system should find the lamp at around $-0.05°$ elevation. If we compare this value to the values given in Table 2, we can conclude that the instrument sees the lamp close to the expected position. The small deviations between the table values and the expected elevation can be explained by a combination of several small uncertainties, namely, the minimum motor step size of $0.01°$, the used scan

resolution of 0.02° and the uncertainties of the calculation of the lamp position $\widehat{\alpha}$ using Eq. (1). Further, also the determination of the expected lamp elevation has an uncertainty as outlined above.

The relatively small span of lamp positions obtained on different days and at multiple wavelengths indicates that this method is very stable and reproducible. Furthermore, this approach allows the calibration of several instruments with the same setup at the same time, since all instruments can point at the same target. However, depending on the slight horizontal distances of the different measurement locations, small differences in the azimuth angle (up to 1.8° for the CINDI-2 campaign) under which the lamp can be seen have to be taken into account. A drawback of this method is that the position of the artificial light source relative to the instrument has to be determined accurately, which might be challenging or even impossible at some locations. Also finding a suitable location for the lamp can be difficult e.g. in cities.

## 3.3 Near lamp measurements

### 3.3.1 Approach

This method also uses an artificial light (Hg-lamp) source during nighttime, but here it is located rather close to the instrument's telescope (a few meters). In order to determine the expected lamp position, namely $\beta = 0°$, the light source has to be aligned to the (center of the) telescope unit of the instrument. This alignment is typically done using a laser level which illuminates both the instrument and the position of the lamp. The telescope and the lamp are then centered around the position of the laser beam. The leveling accuracy of the laser level which was used during CINDI-2 was tested in the laboratory and amounts to approximately 0.1°. Further, the laser beam has a thickness of about 2 mm which translates to another 0.04° uncertainty of the relative vertical positions between instrument and lamp. Both the setup and the alignment procedure are sketched in Fig. 6. Using this procedure, the light source should be found at 0° elevation and possible deviations from that position can be used to correct the elevation calibration.

Such near lamp measurements were not performed for the MPIC instrument during the CINDI-2 campaign. However, the elevation angles of the IUP-HD instrument were calibrated using this method. Therefore, in the following, data from the IUP-HD instrument are used to illustrate this method and its analysis in more detail.

### 3.3.2 Results for the IUP-HD instrument

Three such near lamp scans were done by the IUP-HD group in one night in the preparation phase of the CINDI-2 campaign. Mean intensities are calculated separately for the UV and VIS spectrometer. It should be noted that the first two scans were performed in an elevation range from −2° to 2°, while the last was done in a range from −1° to 0.45°. Since the elevation pointing is continuously regulated by comparison of the orientation of the telescope measured by the built-in tilt sensor with

the nominal angle, no backlash effects are expected. In order to analyse these measurements, the (normalised) mean intensities are plotted against the elevation angle $\alpha$ reported by the measuring system. Next, a Gaussian function of the form

$$S(\alpha) = A + B \cdot \exp\left(-\frac{(\alpha - \widehat{\alpha})^2}{2 \cdot \sigma^2}\right) \tag{2}$$

is fitted to the intensities and the center $\widehat{\alpha}$ of this function represents the lamp elevation. Further, $S(\alpha)$ represents the fitted intensity at a given elevation, $A$ represents an intensity offset, while $B$ describes the maximum of the fitted curve. The width of the fitted curve is controlled by the parameter $\sigma$. For improving the statistics, all three scans are plotted in one plot (using different colours for the individual scans) and the Gaussian fit is applied to the whole data set of one spectrometer (Fig. 7).

The retrieved lamp elevation is also shown in this figure. Following this procedure, lamp elevations $\widehat{\alpha}$ of −0.14° and −0.11° were found in the UV and visible spectral range, respectively. These lamp elevations can be now used to adjust the initial elevation calibration of the instrument.

### 3.4 Horizon scans

#### 3.4.1 Approach

A common method for the calibration of the elevation angles of MAX-DOAS systems is the so-called horizon scan. Here, the elevation $\beta$ of the visible horizon, which is defined as the transition of the tree tops to the open sky, is used as reference. Since this method does not require an active light source, it can be performed during daytime and the variation in the measured intensity at the horizon is used to determine its position. A Gaussian integral is fitted to the measured intensities and the fit parameters give the horizon position. In practice, sometimes the numerical derivative of the intensity curve is calculated since below the horizon the intensity does not approach zero but the rapid change of the measured intensity allows the identification of the horizon position $\widehat{\alpha}$. Prerequisites of this method (despite the knowledge of the expected elevation of the visible horizon) are high visibility, the absence of rapidly varying and/or low-lying clouds and a clear and rapid change in intensity at the visible horizon, which might not be fulfilled during episodes of fog, when the horizon might be blurred. If these conditions are not fulfilled, no clear conclusions can be drawn from horizon scans. Furthermore, it should be noted that the visible horizon should not be too far away (less than a few kilometers) to minimise the influence of atmospheric scattering.

During the CINDI-2 campaign, horizon scans were included in the measurement protocol in order to study the consistency and stability of the elevation calibration of the different measurement systems. Thus, all MAX-DOAS instruments (both 1D and 2D) performed horizon scans between 11:40 UTC and 11:45 UTC at a specified total integration time of 5 s while pointing in the main viewing direction (287° azimuth angle). The scans were done using predefined elevation angles between −5° and 5°, whereby the scan resolution was 0.2° in the interval between −2° and 2° and 1° outside this range.

### 3.4.2 Results for the MPIC instrument

For the MPIC instrument, horizon scan data is available starting from 17[th] September until 2[nd] October. Before 17[th] September some horizon scans were performed as well, but they are of limited quality due to an error in the measurement script of the MPIC system. Furthermore, some days are not used either due to bad weather conditions with fog and many low clouds or due to known pointing problems. On overall 10 days, useful horizon scan data are available for the MPIC instrument.

First, the measured intensity is normalised with respect to the total integration time. As a second step, the intensity curves are also normalised to their corresponding maximum allowing a direct comparison of the intensity curves recorded on different days with various sky conditions. The normalised intensity curves obtained at 340 nm for the different days are shown in Fig. 8(a) (coloured dots). Here, the increase of the measured intensity around the horizon is clearly visible in an elevation range from around 0° to 1°. Next, a Gaussian integral of the form

$$S(\alpha) = A \cdot \left( \mathrm{erf}\left( \frac{\alpha - \widehat{\alpha}}{B} \right) + 1 \right) + C \cdot (\alpha - \widehat{\alpha}) + D \tag{3}$$

is fitted to the data since this approach is more stable than calculating a numerical derivative. Here, $S$ represents the fitted intensity, $\alpha$ the elevation angle and the parameters $A$, $B$, $C$ and $D$ determine the exact form of the fitted curve. The parameter $\widehat{\alpha}$ indicates the center of the fitted function and therefore represents the derived horizon elevation. The resulting daily fit functions are also displayed in Fig. 8(a) by lines in the corresponding colours.

Additionally, the analytical derivative of Eq. (3) can be calculated. The resulting curves which are discplayed in Fig. 8(b) contain information on the instrument's field of view (FOV) since the full width at half maximum (FWHM), which is a typical measure for the FOV, can be derived:

$$FWHM_{Integral} = 2\sqrt{\ln 2}\, B. \tag{4}$$

By following this procedure, a value of 0.30° is found as the median center (vertical red line in Fig. 8) for the fitted functions representing the median horizon elevation for the MPIC instrument at 340 nm. However, it should be mentioned that there is quite some scatter in the daily horizon scans, which might be caused by varying sky conditions on the different days and is one of the drawbacks of this method. The same procedure is also applied to the intensities recorded at 440 nm in order to study possible wavelength dependencies, the resulting intensity curves and derivatives are shown in Fig. 8(c) and (d). Here, a median horizon elevation of 0.37° is obtained, which is slightly higher than the value for 340 nm. These two wavelengths were chosen for the analysis because they were reported by all instruments that participated in the semi-blind inter-comparison during the campaign and thus, they are well suited for a comparison of the horizon scan results for different instruments which is performed in Sect. 4.2.

### 3.5 Bright/white stripe scans

#### 3.5.1 Approach

The white stripe method can also be applied under daylight conditions and a white or at least bright stripe in front of a black/dark background is used as reference target. In order to calibrate the elevation angles, the (center of the) white stripe has to be aligned with the (center of the) telescope, archiving an expected stripe position of $\beta = 0°$. This can be done by using a water or laser level.

 The setup applied by MPIC during the CINDI-2 campaign used an adjustable white stripe in front of a dark plate and a large water level which consisted of two bottles of water which were connected via a 10 m long tube filled with water and positioned next to the stripe and the telescope. On the telescope side the water level has to be adjusted to the middle of the telescope, thus on the plate stripe side the water level indicates the altitude of the telescope. Here, the stripe has to be adjusted to the water level position which guarantees the alignment of stripe and telescope axis. A sketch of the described setup can be found in Fig. 9. The horizontal distance between the telescope and the white stripe was 342 cm and the vertical extension of the stripe was around 2.5 cm which corresponds to a FOV of around 0.4°. This apparent FOV is quite large and shows that the setup was not optimised but the rather short distance between telescope and stripe was determined by the local conditions (a water channel in front of the instrument container limited the maximum distance which could be achieved). Therefore, this calibration method using the here described setup was applied only once during the campaign and only for the MPIC instrument. However, other groups (e.g. BIRA) applied the same method using their own setups. The scan resolution was 0.05° which was a compromise between speed (needed because of the unstable setup) and accuracy. In the following, the analysis is done for two wavelengths, namely 340 nm and 440 nm in order to be consistent with the approach described in Sect. 3.4.

#### 3.5.2 Results for the MPIC instrument

Again, the recorded intensities are first normalised with respect to their total integration time. Next, a background correction is applied to the intensity curves, which is needed since the dark background of the white stripe does not absorb all incident light. For that, a second order polynomial is fitted to the background intensities and subtracted from the measured intensities. The resulting intensity curve at 440 nm and the fitted Gaussian function (compare Eq. (2)) are depicted in Fig. 10. Now, the center $\widehat{\alpha}$ of the Gaussian fit indicates the stripe position. In that way, a value of −0.01° is found (red dashed line). Since the intensity curve again shows no smooth behavior (see Sect. 3.2), additionally the center of mass approach following Eq. (1) is applied, yielding a stripe position of −0.02° (blue dashed line) consistent with the Gaussian approach. Conducting the same procedure for the intensities measured at 340 nm yields values of 0.02° and 0.00° for the Gaussian and center of mass approaches, respectively.

 In summary, a range of −0.02° to 0.02° for the retrieved stripe positions is obtained, which corresponds to only four motor steps. Similarly to the far lamp measurements (Sect. 3.2) this range can be explained by the minimum motor step size of 0.01°, the used scan resolution of 0.05° and the uncertainties of the retrieval of the stripe position $\widehat{\alpha}$. Further, an error of ±5 mm in the

alignment between telescope and stripe was estimated which translates to an uncertainty of ±0.08°. Finally, also the angular height (0.4°) of the white stripe was quite large.

## 3.6 Sun scans

Sun scans use the solar disc as reference target. Its position is a function of time and the geo location (longitude and latitude) of
the instrument and thus, the expected position of the sun can be calculated with high accuracy. The position of the solar disc is described by two angles, namely the elevation angle and the azimuth angle. Therefore, this method can only be applied to 2D instruments which allow scans in these two directions. In order to determine the elevation angle $\widehat{\alpha}$ (and the azimuth angle) under which the solar disc can be found, a scan across the solar disc (and its surrounding) similar to the methods described previously is performed. The retrieved value for $\widehat{\alpha}$ which was determined in the a-priori elevation calibration, is then compared to the
calculated elevation $\beta$ and possible deviations can be corrected. Since also the azimuth angle of the sun can be determined, this method is also well-suited to calibrate the azimuth angles of 2D MAX-DOAS systems. This topic, however, is not part of this study and therefore not investigated in more detail here.

During the CINDI-2 campaign several 2D instruments, e.g. BIRA, AUTH and the PANDORA instruments operated by KNMI, LuftBlick and NASA, performed sun scans to calibrate their elevation (and azimuth) angles. The PANDORA systems
even performed sun scans on a regular basis to actively monitor and adjust their elevation (and azimuth) calibrations. As shown later, the results for these instruments show good agreement to the other presented calibration methods. Advantages and disadvantages of this method will be discussed in the conclusion section (Sect. 5) of this paper.

## 3.7 Comparison between the different methods

In this section, the different methods for the elevation calibration of MAX-DOAS instruments are compared and examined with
respect to their consistency. Table 3 summarises the retrieved target elevations $\widehat{\alpha}$ using the methods introduced above (except the near lamp measurements and the sun scans which were not done for the MPIC instrument).

A comparison of the results for the far lamp measurements and the white stripe scans shows very good agreement, with small deviations that can be explained by the already mentioned motor step size of 0.01°, the different scan resolutions and the uncertainties related to the retrieval of the apparent target positions $\widehat{\alpha}$. For both methods, no significant wavelength dependence
of the retrieved target positions is visible.

However, comparing the results for these two consistent methods to the results of the horizon scans shows larger deviations. The retrieved median horizon positions from the horizon scans for 340 nm and 440 nm were 0.30° and 0.37°, respectively, which are significantly higher than the values around 0° found with the other two methods. During the campaign, the xenon lamp was placed directly in front of a row of trees which mark the visible horizon (the transition of the tree tops to the open sky).
As already explained in Sect. 3.2, the vertical position of the lamp was 3.5 m above the water level in the water channel which was located next to the measurement site (see Fig. 3), and a few meters below the tree tops. Thus, during the day the visible horizon is not represented by the ground but by the tree tops situated close to the lamp location in the main viewing direction of the MAX-DOAS instruments. Converting the retrieved horizon elevations of 0.30° and 0.37° (for 340 nm and 440 nm) into

vertical distances at lamp location using geometry leads to heights of 6.7 m and 8.3 m, respectively. The mean retrieved lamp position is 0.01° which corresponds to 0.22 m at lamp distance and therefore, the visible horizon is seen 6.5 m and 8 m above the altitude of the lamp for 340 nm and 440 nm, respectively. These values are in the order of typical tree heights and therefore, these calculations can explain the deviations (of about 0.3° to 0.37°) to the other two methods. Later this information is used to estimate the expected horizon elevation for the other instruments.

A further effect which can be observed for the horizon scans is that the median horizon elevation is significantly lower for 340 nm compared to the one obtained at 440 nm. As we will see in the next section, this effect is also found for the other participating MAX-DOAS systems. Besides this systematic wavelength dependency of the horizon scans, the results of the different methods agree quite well amongst each other when taking the uncertainties into account.

## 4 Comparison between different instruments

In this section the results of the far lamp measurements and the horizon scans are inter-compared between the different participating groups.

### 4.1 Comparison of the far lamp measurements

Far lamp measurements using the xenon lamp as described in Sect. 3.2 were performed in several nights throughout the campaign. However, not all instruments participated each time. Since the different instruments use rather different instrumental setups and scanning schemes, they are divided into three groups.

#### 4.1.1 Full 2D scans

The first group consists of 2D instruments which performed full 2D scans of the xenon lamp in vertical (elevation angle) and horizontal (azimuth angle) direction on at least one night. For these instruments, the measured intensities are first normalised with respect to integration time and interpolated to specific wavelengths in order to compare the results of the different methods and instruments. Column (a) of Fig. 11 shows representative examples of the obtained 2D intensity distributions for the BIRA, IUP-HD, UToronto and LMU (for this instrument only the mean intensities of the spectra are available) instruments, respectively. Additionally, black dotted lines indicating the azimuth angle under which the maximum intensity was recorded can be found in these figures. The axes of these sub-figures were chosen in a way that they all show the same relative elevation (1°) and azimuth span (1.2°). While the BIRA instrument shows a very smooth and smeared out distribution of the measured intensities, the intensity distributions are more sharp for the UToronto (still quite smooth), IUP-HD, and LMU instruments. This finding can be explained by the fibre configurations inside the telescope units of the four instruments since they have an influence on the actual shape of the measured intensity distributions. While the LMU and IUP-HD instruments used a ring of fibres inside their telescope units (for the UV channel), the UToronto and the BIRA UV instruments used a spot configuration, consisting of 37 and 51 fibres, respectively. When scanning across the xenon lamp, it might occur that the FOV is not always fully illuminated at the "edges" of the xenon lamp light beam. The ring configuration might be more sensitive (similarly to

the fibre effect found for the MPIC instrument in Sect. 3.2) to this effect and introduce more edges to the measured 2D intensity distributions, leading to a sharper shape. Further, also differences in the motor pointing precisions have an effect on the apparent FOVs.

Two approaches were applied to retrieve the horizontal position under which the lamp is found for this first group of instruments. For the first approach, the intensities along a transect (black dotted lines in column (a)) are extracted and a Gaussian function (Eq. (2)) is fitted to these intensities. The centers $\widehat{\alpha}$ of these fits represent the lamp elevations, the intensity curves and Gaussian fits for the four examples can be found in column (b) of Fig. 11. For the second approach, all intensities which were recorded at one specific elevation angle are integrated over the different azimuth angles. These values are then used for the analysis and again, the center of a Gaussian fit indicates the vertical position of the light source. Column (c) of Fig. 11 depicts the resulting curves and fits for the four instruments. The results of the two methods are very consistent for a single instrument. Nevertheless, there is quite some spread between the different instruments, which will be investigated in more detail at the end of this section.

### 4.1.2   Cross-scans

The second group are 2D instruments which performed cross-scans, meaning that first an azimuth scan was performed, followed by an elevation scan at the azimuth direction under which the maximum intensity was found. This was done by three instruments using individual scanning schemes. Examples of the obtained intensity curves and corresponding Gaussian fits are depicted in Figure 12. The different panels of this figure show the curves, fits and resulting centers for the AUTH (a), BOKU (b) and IUB-B (c) instruments, respectively. The results for the lamp position are rather consistent for the different scans for an individual instrument since the obtained centers are nearly the same. This is also valid when looking at the results for different wavelengths for one instrument (not shown here). However, it can be seen that there is some spread between the different instruments.

### 4.1.3   1D scans

The last group consists of 1D instruments which performed simple elevation scans of the xenon lamp as described in Sect. 3.2. For these instruments, the normalised intensity is plotted against the elevation angle and the center of a Gaussian fit gives the lamp elevation. Examples for the CMA UV, CMA VIS, BSU and AIOFM instruments are shown in Fig. 13, with the resulting lamp elevations (centers) also displayed. Since the BSU instrument has a 2D CCD on which the second dimension represents the elevation angle, it should be noted that this instrument did not really scan across the lamp but each image on the CCD represents a full lamp scan. The AIOFM instrument is a 2D instrument, but was operated in 1D mode for the far lamp measurements.

### 4.1.4 Analysis of the far lamp scans

For each participating instrument, the intensity curves are extracted for all valid lamp measurements by applying the respective procedure explained above for different wavelengths (365 nm, 405 nm, 436 nm and 546 nm) corresponding to the individual spectral ranges of the instruments. Further, a Gaussian function (Eq. (2)) is fitted to the data. The fit parameters are initialised by $A_0 = 0$, $B_0 = $ maximum of the measured intensity curve, $\widehat{\alpha}_0 = $ center of mass (calculated using Eq. (1)) and $\sigma_0 = 0.5°$.

The resulting lamp elevations are summarised in Fig. 14, where the mean of all retrieved lamp elevations (at different wavelengths and/or scans/days) for each instrument is shown as dots. Further, for each instrument, three different measures for the uncertainties of the retrieved lamp elevations are displayed. The left error bar of each instrument indicates the mean of all fit errors of the fits explained above and measures the quality of the individual fits and the shape of the measured curves. The standard error of the mean of all retrieved lamp elevations is represented by the middle error bar. It is a measure of the consistency and stability of the results of the different lamp scans performed by one instrument. However, this quantity also depends on the actual number of available intensity curves at different wavelengths and days which is given in brackets behind the institute acronyms on the x-axis in Fig. 14. Lastly, the right error bar indicates the daily spread which is only available for instruments which performed more than one scan on one day and for all 2D instruments, since two methods were applied to extract the 1D intensity curves. The daily spread of one day is defined as the standard deviation of the results of the different scans on that day. If in addition several days are available, the mean of the daily standard deviations is calculated and displayed.

As shown in Fig. 14, a rather high spread of around 0.9° is found for the retrieved lamp elevations. Nevertheless, the values are centered around the expected values of −0.19° (dashed blue line) and −0.05° (dashed green line) for the instruments located on the upper (mostly 2D instruments) and lower (mostly 1D instruments) row of containers installed at Cabauw, respectively. These expected values were calculated as described in Sect. 3.2, where a $\Delta h$ of 1 m was estimated for the instruments located on the lower row of containers (Fig. 3). The instruments on the second row of containers are placed around 3 m higher than the instruments on the lower row and therefore, the same calculations yield an expected lamp position of −0.19°. Further, most of the error bars for the individual instruments are quite small, indicating the good stability and repeatability of the far lamp measurements. The large error bar for the mean fit error for the LMU instrument can be explained by a rather uneven intensity distribution which leads to bad fit results in some cases.

The deviations between the different instruments are on the one hand caused by slightly different vertical positions (even if they are located on the same container level) of the instruments, since some of the instruments were mounted on tripods or similar devices while other instruments were placed closer to the container roof. On the other hand, the deviations are also caused by the fact that all groups reported their elevation angles corresponding to their own elevation calibrations. Therefore, the spread of about 0.9° of the retrieved lamp elevations (for one container level) is a measure of variability between the elevation calibrations of the different instruments.

More details will be discussed in Sect. 4.3, where the derived lamp elevations are compared to the corresponding horizon elevations obtained from the daily horizon scans which are inter-compared in the next section.

## 4.2 Comparison of the horizon scans

As mentioned above, during the day the visible horizon might be defined by obstacles such as trees. Like explained in Sect. 3.7 the horizon elevations of the MPIC instrument were used to estimate the height of the visible horizon above the lamp altitude yielding values of 6.5 m and 8 m at 340 nm and 440 nm, respectively. Taking into account the expected lamp elevations which are slightly below 0° (Sect. 4.1), we estimate the elevation of the visible horizon to be 0.24° at 340 nm and 0.31° at 440 nm for the instruments on the lower row of containers. However, since some of the other instruments (mainly the 2D ones) were located on the second row of containers, which was around 3 m above the lower row, the expected horizon elevation for these instruments is lower and we derive elevations of 0.10° and 0.17° if we use the same assumptions. As we will see below, the difference between the estimated horizon elevations for the two rows is smaller than the spread of the results of the horizon scans between the individual instruments.

In the following we use the results of the horizon scans of the participating instruments derived from the reported intensities of the daily horizon scans at 340 nm and 440 nm following the approach explained in Sect. 3.4 and Kreher et al. (2019).

First, median horizon elevations are calculated for both wavelengths for all instruments. These median values (dots) are depicted in Fig. 15 together with the corresponding 25 % and 75 % percentiles (lines) which are a measure of the spread of the daily horizon elevations. No percentiles are shown for the NIWA_EnviMeS instrument since horizon scans were reported only for one single day. The values at 340 nm are represented by blue dots and lines, while red dots and lines indicate the values at 440 nm. Additionally, the expected horizon elevations at 340 nm and 440 nm for the instruments on the upper container row are displayed as blue and red dotted lines, respectively. It should be mentioned that only results for instruments which reported valid horizon scans on at least one day are shown.

Obviously, the retrieved horizon elevations of the different instruments have a quite large scatter of around 0.9° which is consistent with the spread of the lamp elevations that were discussed in the previous section. Nevertheless, the results for many instruments are centered around the estimated values from above. Especially, the results of the Pandora systems (operated by LuftBlick and NASA) which used sun scans to calibrate the elevation angles are very close to that value. As indicated previously, the calculated difference of around 0.14° between upper and lower row in the elevation of the visible horizon is insignificant compared to the rather large scatter between the individual instruments. Even calculating mean or median horizon elevations for the instruments located on the upper and lower row which are displayed in Table 4 does not reveal this expected difference. However, when excluding some obvious outliers in the calculation of the mean (or median) elevations, there is some indication of this effect which would be, however, rather arbitrary.

Looking at the percentiles of the individual instruments shows that for many of them the results of the daily horizon scans have a high degree of consistency which is indicated by the rather small percentile lines in Fig. 15. However, some instruments (e.g. the CMA instruments, the AUTH or the LMU instruments) show quite large percentile lines, indicating more variable results of the daily horizon scans. This finding can also be seen in Fig. 16 and 17, where the results of the horizon scans at 340 nm and 440 nm are displayed on a daily basis for the individual instruments (blue dots), together with the corresponding median (blue dashed lines) and expected (black dashed lines) horizon elevations. The instruments having small percentile lines

show a smooth behaviour in the daily horizon elevations, while the ones having larger percentile lines show a higher dispersion (e.g. the CMA_UV instrument). Some of the instruments (e.g. the LMU instrument) show a systematic behaviour in the daily results which might indicate adjustments of the elevation calibrations of these instruments, especially in the beginning of the campaign.

The rather large scatter between the instruments has several reasons. First, the individual instruments are placed at slightly different heights due to their specific instrumental setup even if they are located on the same container level. Further, the horizon is not a smooth line and the exact horizon position depends on the exact azimuth pointing direction and the actual shape of the horizontal line. Also limited accuracy of the initial elevation calibrations of some instruments might contribute to this rather high dispersion of the horizon elevations. Lastly, also the weather (especially clouds) and visibility have a significant influence

on the results of the horizon scans since they have an impact on the actual intensity variation around the visible horizon.

Another finding is that the horizon at 340 nm is found at significantly lower elevations compared to 440 nm for most instruments which reported both intensities. However, it should be mentioned that for some instruments (e.g. the IUP-HD instrument), the different wavelength ranges are recorded using separated sensors and even separate optical entrance systems. Nevertheless, these instruments show the same systematic difference. This finding is consistent with the results obtained with

the MPIC instrument but so far no completely clear explanation could be found. However, an investigation of the intensity curves at different wavelengths for the horizon scan performed on one day shows an unexpected intensity variation at $0°$ elevation. An example is shown in panel (a) of Fig. 18 which displays the normalised intensity curves at different wavelengths measured on $17^{\text{th}}$ September (similar behaviours are found on other days). Here, a local minimum is visible at $0°$ elevation for wavelengths above 370 nm, which gets more pronounced for increasing wavelengths. This minimum influences the Gaussian

integral fits (dashed lines in 18(a)) and the retrieved horizon elevations $\widehat{\alpha}$ show a quite smooth wavelength dependence as depicted in panel (b) of Fig. 18. These findings indicate a surface albedo effect. Pictures from the site show that in the visible spectral range the trees close to the horizon appear darker than the grass below them. This can be explained by the fact that the sun altitude is quite high during the horizon scans and the trees are illuminated at a rather flat angle, while the grass is illuminated in a very steep angle. In that way, the local minimum at $0°$ can be explained. However, it is not clear why at shorter

wavelengths no such increase of the measured signal towards smaller elevation angles is found. Nevertheless, measurements which were collected during the first CINDI campaign support these indications for a surface albedo effect. Lastly, it should be mentioned that radiative transfer simulations showed that the wavelength dependence of atmospheric scattering cannot explain the observed wavelength dependence of the horizon scans. This unexpected wavelength dependence of the elevation scans is another drawback of the horizon scans and if possible they should be analysed in the UV spectral range (at wavelengths below

ca. 370 nm).

## 4.3   Comparison between far lamp and horizon elevations

Finally, the results for the far lamp measurements and horizon scans are compared for all instruments which recorded far lamp spectra and performed a horizon scan on at least one day. For that, the mean horizon elevations are plotted against the corresponding mean lamp elevations. This plot is done separately for the horizon elevations retrieved at 340 nm and 440 nm since

a systematic difference for the results at the two wavelengths was found for the horizon scans. However, no such dependency was found for the lamp measurements. The resulting correlation plots can be found in Fig. 19 and Fig. 20, respectively. The error bars in the x- and y-direction indicate the standard errors of the mean retrieved lamp and horizon elevations, respectively.

A first finding is that the error bars are larger for the horizon scans for most of the instruments. On the one hand, this can be explained by the larger number of horizon scan measurements since most of the groups performed lamp measurements only on one or two days. On the other hand, however, the horizon scans are more sensitive to the different weather/sky conditions which influence the exact position of the visible horizon. Furthermore, the horizon is not a homogenous line and therefore uncertainties arise from the uncertainty of the exact azimuth angles of the different instruments. Lastly, for many of the instruments the horizon scans are performed at coarser resolution compared to the lamp scans, which also might effect the results and the resulting uncertainties.

In order to examine the consistency between the two methods, a total least squares (TLS) fit (Cantrell, 2008) is applied to the data. This fitting method takes the uncertainties of the obtained values in both x- and y-direction into account by weighting the values with respect to their uncertainties. In that way values with a smaller uncertainty get more weight compared to those with a larger uncertainty. Here, the standard errors of the mean horizon and lamp elevations were used as measure of uncertainty. The resulting fit lines (solid red lines) with their corresponding slopes of 1.17 and 1.10 and their intercepts of 0.32° and 0.40° for 340 nm and 440 nm, respectively, are also displayed with their fit errors in the corresponding figures as well as the 1:1 line (dashed red line).

Taking all the uncertainties of the two different methods into account the slopes obtained by the TLS fits are close to 1 and therefore, we can state that the results of the two methods correlate well. Nevertheless, the horizon is consistently seen at around 0.2° to 0.6° (except the CMA UV instrument which has a very large error bar for the horizon scans) higher elevations compared to the lamp scans which corresponds to around 4.5 m to 13 m at lamp distance and is in agreement with the calculations in Sect. 3.7. As already described there, this systematic offset between horizon and lamp scans can be explained by the fact that during daytime the visible horizon is represented by trees. Further, this consistent offset between the two methods is represented by the intercepts of the TLS fits (0.32° and 0.40°). Finally, the difference of 0.1° in the offsets obtained for the two wavelengths is consistent with the findings from Sections 3.4 and 4.2.

All in all the two methods presented here yield consistent results for most of the instruments and therefore both are suited to calibrate the elevation angles of MAX-DOAS systems.

## 4.4 FOV determination

All presented methods use elevation scans across reference targets and the apparent elevation angles $\widehat{\alpha}$ under which the targets can be found are determined by fitting either a Gaussian function or a Gaussian integral. These fits contain also information on the instruments' fields of view (FOV) which are represented by the full widths at half maximum (FWHM). For the Gaussian

integral which was used for the horizon scans the FWHM can be determined using Eq. (4). The FWHM for the Gaussian fit can be calculated using

$$FWHM_{Gauss} = 2\sqrt{2\ln 2}\,\sigma, \tag{5}$$

where $\sigma$ is the parameter controlling the width of the Gaussian function (Eq. (2)) which is fitted as explained above. For the
instruments which performed both far lamp and horizon scans, the FOVs were determined as outlined and compared to their reference FOVs which were measured in the laboratory prior to the campaign and are listed in Table 1. The FOVs derived from the horizon scans and far lamp measurements were plotted against the corresponding reference FOVs in Fig. 21. Here, blue dots indicate FOVs obtained from the far lamp measurements, while green dots show the results retrieved from the horizon scans. Further, linear regressions (blue and green dashed lines) with their corresponding slopes and a 1:1 line (red dashed line)
are shown. It should be noted that the results for the UToronto (not shown at all) and the AIOFM (pale blue dot in panel (a)) instruments were not included in the regressions since their retrieved FOVs are obviously problematic.

     Both panels of Fig. 21 show that the results are not as clear as in case of the determination of the horizon and lamp positions. This indicates that the centers which were retrieved so far, are more robust and less sensitive to the specific shape of the recorded intensity curves. Nevertheless, the obtained slopes of the regression lines are not that far from 1 which indicates that
in principle the retrieval of the FOV from the elevation calibration methods is possible. However, especially in the visible spectral range there seems to be a systematic underestimation of the retrieved FOVs as compared to the reference FOVs for most of the instruments. The reason for this is so far not clear and would need a more sophisticated assessment. This is, however, outside the scope of this paper since focus of this paper is put on the elevation angle calibration. Another finding is that systematically slightly larger FOVs are found for the horizon scans compared to the lamp scans, especially in the UV
spectral range. This can (at least for the 1D instruments) be explained by the fact that the far lamp was not always in the center of the azimuth dimension of the FOV for some instruments.

## 5   Overall discussion and conclusions

If the optical axis of a MAX-DOAS instrument would be precisely known, a simple water level could be used to calibrate the elevation angles. This assumption, however, is not always fulfilled, e.g. because the fibre bundle is not perfectly centered.
Because of these uncertainties the in-field calibration of the elevation angles as described in this paper are indispensable.

     Five different methods for the calibration of elevations angles of MAX-DOAS instruments were described and discussed. All were applied during the CINDI-2 campaign by at least one group. The approaches of three methods were explained and examined using data from the MPIC MAX-DOAS instrument, while the near lamp measurements were described using data from the IUP-HD instrument. The white stripe measurements, using the MPIC setup, were only done for the MPIC instrument.
Finally, the sun scans were described. The results of the horizon scans and the far lamp measurements, were inter-compared quantitatively between the individual groups and amongst each other.

All of the methods use the same principle. First, a specific target is placed in front of the telescope unit of a MAX-DOAS system. This target has to be located at a fixed and known location relative to the optical unit of the instrument. In that way, an expected elevation angle $\beta$ under which the target should be found can be determined. Then the apparent elevation angle $\widehat{\alpha}$ of the target is retrieved by the MAX-DOAS system using the measured intensities of an elevation scan across the target. Finally, the elevation angles of the instrument can be calibrated by comparing the expected and retrieved target positions and adjusting the telescope position in an appropriate way.

Tables A1 and A2 list systematic and statistical effects, respectively, which introduce uncertainties to the measurements and analyses presented in this paper. In summary it can be concluded that the systematic effects which determine the accuracy of a method are dominated by the uncertainties introduced during the determination of the target positions and the fit errors of the retrieval of the apparent target positions. The statistical errors which limit the precision of a method, however, are dominated by motor uncertainties. These are characteristic properties of the used motors/steppers of the individual instruments and therefore, the precision of the methods is prescribed by the individual properties of the motors/steppers (typically $\pm 0.02°$ to $\pm 0.1°$). In the following discussion the typical accuracy of each method is estimated by quadratically adding the (systematical) error related to the determination of the target position and the error associated with the Gaussian fits and then applying the square root. For the latter the values are ranging from $\pm 0.01°$ to $\pm 0.1°$ depending on the instrument and the symmetry of the recorded intensity curves. For that reason a value of $\pm 0.05°$ is used in the following. However, it should be noted that for good performing instruments this value can be considerably lower and in many cases the accuracy of a method is simply limited by the uncertainties in the determination of the target position.

The first method used far lamp measurements during nighttime, where an artificial light source close to the visible horizon was used as the target. The results for the MPIC instrument illustrate the stability and reproducibility of this method since similar results are obtained for the different days at multiple wavelengths. The results of the other groups confirm this finding since a rather small spread is found for each of the individual instruments. However, the spread of around $0.9°$ between all instruments is quite large, which is likely caused by the fact that the individual initial elevation calibrations are not done in a consistent way. As described in Sect. 3.2 the uncertainty associated with the determination of relative position of the far lamp is estimated to be $\pm 0.5\,\text{m}$ which translates to $\pm 0.02°$ at lamp distance. Combining this with the fit errors of the Gaussian fits yields a systematic uncertainty of the far lamp measurements of $\pm 0.05°$. All in all, the main advantages of this method are the stability, reproducibility and the possibility to calibrate several instruments with the same setup. Furthermore, this method is very accurate as long as the instrument has a mostly symmetric FOV. For many instruments, the uncertainty is even smaller and is finally limited by the accuracy and precision of the motors of the instruments. Nevertheless, it might be difficult to find suitable lamp locations and to determine the position of the lamp relative to the telescope for most measurement sites. Therefore, the overall effort for this method is quite high. Lastly, this method can be applied only during nighttime. It should be mentioned that this method can also be used for calibrating the azimuth angles of 2D instruments since also the azimuthal direction of the lamp can be determined, however, it might be difficult to find suitable lamp locations at different (azimuthal) locations.

Near lamp measurements offer another way to perform an elevation calibration. This method is very similar to the far lamp measurements, but here the lamp is located rather close to the instrument. Following the procedure above and using the uncertainties as outlined in Sect. 3.3 the systematic uncertainty of this method is estimated to be around main ±0.1°. Like for the far lamp measurements the main advantages are accuracy and stability. Furthermore, the setup is more compact and requires less effort, especially in determining the position of the lamp relative to the telescope. However, when comparing the typical accuracy of the near and far lamp measurements, the distance between the lamp and the telescope is quite small for the near lamp measurements. This leads to a rather large angular fraction which is occupied by the lamp and influences the results of the lamp scan. Finally, this method can also be applied only during nighttime and only for one single instrument at once using the same setup.

Horizon scans during daytime use the visible horizon as target. Consistent results were found for all participating instruments compared to the far lamp measurements and also here a spread of 0.9° between the individual instruments is found. Further, these consistent results are represented by a good correlation of the retrieved apparent horizon and lamp elevations. Here, the obtained slope is close to 1 and the obtained intercepts of 0.32° and 0.40° indicate an offset between lamp and horizon measurements. This offset can be explained by the fact that the visible horizon differed from the real horizon since trees represented the visible horizon during day. By knowing the height of these obstacles, this problem can be solved and the offset between the method can be explained as examined in Sect. 3.7 and 4.3. Additionally, it should be emphasised that the results of the horizon scans show a wavelength dependence (higher horizon elevations for larger wavelengths) which is likely caused by surface albedo effects and needs further investigation. Although the spread between the results for the different instruments is of the same order as for the far lamp measurements, the spread for the individual instruments is significantly larger than for the lamp measurements. This is caused by the large sensitivity of the horizon scans to varying weather conditions and visibility. Additionally, more horizon scans were performed, which naturally increases the scatter in the retrieved horizon elevations since the scans were performed with different underlying sky conditions. Nevertheless, this method and its principle are quite simple and can be easily implemented in measurement protocols. Thus, this method can be easily performed on a regular basis and used for monitoring the elevation calibration. Taking all this together, shows that the horizon scans can be a good calibration tool if besides the knowledge of the position of the visible horizon only scans under favourable conditions are used. Hence, only days with good visibility should be used and days with rapidly varying cloud cover and/or low-lying clouds should be avoided. One major drawback is the difficulty to determine the height of the visible horizon e.g. in mountainous regions. Also for comparison exercises it should be mentioned that the horizon is not a homogenous line but might have some inhomogeneities that can influence results for the individual instruments and therefore might introduce some discrepancies between the instruments. For the CINDI-2 campaign the uncertainty of the determination of the position of the visible horizon was estimated as roughly ±5 m. Combining this with the fitting uncertainties yields an systematic uncertainty of the horizon of ±0.25°.

White stripe measurements are another method which can be applied during the daytime. During the CINDI-2 campaign, the setup, which was used by MPIC, was rather experimental and measurements using this setup were performed only by the MPIC instrument. Nevertheless, the results are promising and consistent with the other methods and also other groups (e.g. BIRA)

applied this method utilising a different setup. Since the position of the stripe relative to the optical unit can be determined very accurately, this method is very accurate. However, a stable setup is needed in order to perform a good calibration. For this method, a typical uncertainty on the order of ±0.1° is estimated, when using an error of ±5 mm in the determination of the strip position as explained in Sect. 3.5. But since only one such measurement was performed by the MPIC instrument, further
studies are needed in order to confirm this estimate.

    Finally, sun scans were described and explained. Here, the errors introduced by the determination of the position of the solar disc can be neglected and the systematic uncertainty is dominated by the retrieval of the apparent solar position and the motor properties. This is valid as long as the scans are performed when the sun altitude is large since otherwise atmospheric refraction might play a role. Therefore, we estimate the accuracy of this method to be typically in the order of ±0.05°. Like for
the other methods the statistical uncertainties are given by the motor precision. Main advantages of this method are the high accuracy, the possibility to perform such sun scans regularly and the option to calibrate both elevation and azimuth angles. Major disadvantages of this method are the fact that these scans can only be applied under cloud free conditions and the rather high complexity. Further, this method is only applicable to 2D instruments.

    Table 5 provides a summary of the described methods by listing their setups, the measurements which are needed, their
specific advantages and disadvantages and their typical systematic uncertainties.

    The results of this study show that in principle all of the presented methods can be used for the calibration of the elevation angles of MAX-DOAS systems since they yield consistent results. Nevertheless, the advantages and disadvantages of the individual methods should be considered when deciding for one method. A further finding of this study is the fact that still some improvement in the consistency of the elevation calibration between the different MAX-DOAS systems is needed as
there is still a rather large discrepancy between the different instruments of around 0.9°. This is in the order of a typical FOV of a MAX-DOAS system and might have a significant effect on the retrieved MAX-DOAS trace gas dSCDs. Therefore, for upcoming campaigns strong emphasis should be put on the monitoring and possible correction of the elevation calibration. To do so, if possible and if suitable lamp positions can be found far lamp measurements should be performed at least once by all participating instruments and a common elevation reference (namely the lamp elevation) should be defined prior to a
comparison campaign. In general we recommend that given the high accuracy, 2D instruments should make use of sun scans. Simple 1D instruments should perform near lamp or white stripe scans to accurately calibrate the elevation angles since these two methods are rather accurate and the effort is reasonable. As these two methods don't have a common reference elevation they are more suitable for individual MAX-DOAS stations. Both, during campaigns and for single MAX-DOAS stations, horizon scans (or sun scans for 2D instruments) should be implemented in the measurement protocol and should be performed
on a regular basis in order to monitor the temporal stability of the calibration of the elevation angles.

## Appendix A: Error assessment

This section provides two tables which list and evaluate systematic (Table A1) and statistical errors (Table A2) that influence the measurements described in this paper and therefore cause uncertainties in the elevation calibration.

*Author contributions.* All co-authors contributed to the study either as campaign participants and instrument operators and/or by performing the data analysis, data quality control and/or providing their data of the individual instruments. MVR performed the standardised analysis of the horizon scans. JK, TW and SDO offered and conducted the far lamp measurements during the campaign. Further, they performed the white stripe scans. UF and JLT performed and analysed the near lamp measurements. SDO and TW interpreted the results of the study and wrote the manuscript with feedback and contributions from the other co-authors. TW, SB and SDÖ contributed with valuable feedback and supervised the study.

*Competing interests.* The authors declare that they have no conflict of interest.

*Acknowledgements.* Funding for this study was provided by ESA through the CINDI-2 (ESA Contract No.4000118533/16/I-Sbo) and FRM4DOAS (FRM4DOAS: ESA Contract No. 4000118181/16/I-EF) projects and partly within the EU 7[th] Framework Programme QA4ECV (Grant Agreement no. 607405) project. The BOKU MAX-DOAS instrument was funded and the participation of S. F. Schreier was supported by the Austrian Science Fund (FWF): I 2296-N29.

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

**Table 1.** Overview of the key properties of the instruments participating in the elevation calibration exercise. The table lists the institute names, the corresponding instrument names and their acronyms (columns 1-3). Further, the instrument types (1D/2D) as well as the spectral range and resolution for each instrument are summarised in columns 4 to 6. Additionally, the number of detectors and light couplings, the type of light coupling (fibre F or direct D) and the instrument fields of view (FOV) are specified (columns 7-9). Finally, the corresponding detector temperatures and references explaining the individual instruments in more detail are given (columns 10 and 11).

| Institute | Instrument name | Instrument acronym | Type | Spectral range (nm) | Spectral resolution (nm) | Number of detectors/ couplings | Light coupling | FOV (°) | Detector temperature (°C) | Reference |
|---|---|---|---|---|---|---|---|---|---|---|
| Anhui Institute of Optics and Fine Mechanics | 2D-MAXDOAS | AIOFM | 2D | 290-380 | 0.4 | 1/1 | F | 0.2 | -30 | N/A |
| Aristotle University of Thessaloniki | PHAETHON | AUTH | 2D | 300-450 | 0.4 | 1/1 | F | 1.0 | 5 | Drosoglou et al. (2017) |
| Royal Belgian Institute for Space Aeronomy | 2D-MAXDOAS | BIRA | 2D | 300-390/ 400-560 | 0.37/0.58 | 2/2 | F | 1.0/0.5 | -50/-50 | Clémer et al. (2010) |
| University of Natural Resources and Life Sciences Vienna | 2D-MAXDOAS | BOKU | 2D | 419-553 | 0.8 | 1/1 | F | 0.8 | -60 | Schreier et al. (2019) |
| Belarusian State University | MARSB | BSU | 1D | 300-500 | 0.4 | 1/1 | D | 0.2-1.0 | -40 | Bruchkouski et al. (2014) |
| China Meteorological Administration | MiniDOAS UV | CMA UV | 1D | 300-450 | 0.7 | 1/1 | F | 0.8 | room | N/A |
| China Meteorological Administration | MiniDOAS VIS | CMA VIS | 1D | 400-710 | 1.6 | 1/1 | F | 0.8 | room | N/A |
| University of Bremen | 2D-MAXDOAS | IUP-B | 2D | 305-390/ 406-579 | 0.45/0.83 | 2/1 | F | 1.0 | -35/-30 | Peters et al. (2012) |
| University of Heidelberg | 2D-EnviMeS | IUP-HD | 2D | 300-460/ 440-580 | 0.6/0.5 | 2/1 | F | <0.5 | 20 | Lampel et al. (2015) |
| Ludwig-Maximilians-Universität München | 2D-EnviMeS | LMU | 2D | 300-460/ 450-600 | 0.6/0.9 | 2/1 | F | 0.4 | 20 | Lampel et al. (2015) |
| Max Planck Institute for Chemistry | Tube MAX-DOAS | MPIC | 1D | 305-465 | 0.6 | 1/1 | F | 0.7 | 20 | Donner (2016) |
| University of Toronto | 2D-MAXDOAS | UToronto | 2D | 340-560 | 0.75 | 1/1 | F | 0.62 | -70 | Fraser et al. (2009) |

**Table 2.** Overview of the lamp elevations obtained for all days and different wavelengths for the MPIC instrument. Additionally, the scan resolution is indicated in the last column.

| Date | 365.16 nm | 404.90 nm | 435.96 nm | Resolution |
|---|---|---|---|---|
| 08.09. | −0.15° | −0.16° | −0.19° | 0.10° |
| 10.09. | −0.01° | −0.01° | −0.01° | 0.02° |
| 13.09. | 0.01° | 0.02° | 0.02° | 0.02° |
| 19.09. | 0.01° | 0.02° | 0.02° | 0.02° |

**Table 3.** Overview of the retrieved target elevations $\widehat{\alpha}$ for the MPIC instrument using three of the methods described in the text.

| Method | Target elevation |
|---|---|
| Far lamp | −0.01° to 0.02° |
| Horizon scans | 0.22° to 0.41° |
| White stripe | −0.02° to 0.02° |

**Table 4.** Overview of the mean and median horizon elevations for the instruments located on the upper and lower roof containers and their corresponding standard deviations (STD).

| Container level | Mean (°) | Median (°) | STD (°) |
|---|---|---|---|
| Upper 440 nm | 0.21 | 0.18 | 0.19 |
| Upper 340 nm | 0.14 | 0.11 | 0.17 |
| Lower 440 nm | 0.13 | 0.06 | 0.26 |
| Lower 340 nm | 0.03 | 0.06 | 0.26 |

**Table 5.** Overview on the basic characteristics of the described methods for elevation calibration. The table summarises the basic setup and prerequisites needed (column 2). Column 3 shows properties that have to be measured with their typical (systematic) uncertainties. Note, that the uncertainties in this table are typical values and not the exact ones used in this study. Columns 4 and 5 resume advantages and disadvantages of the different methods, respectively. Finally, the last column gives uncertainty estimates based on the results from CINDI-2 summarised in this paper. Here, it should be noted that also statistical errors occur, which are usually dominated by the uncertainties of the positioners (see Table A2). This uncertainties are usually between 0.02° and 0.1° and are similar for all methods.

| Method | Setup | Measurements needed (and typical uncertainty) | Advantages | Disadvantages | Systematic uncertainty |
|---|---|---|---|---|---|
| Far lamp | Artificial light source at a far distance (around 1 to 2 km) during nighttime | Horizontal distance to lamp (in the order of ±50 m) Vertical lamp position (around ±0.5 m, depending on exact setup) | Accurate and precise Stable and reproducible Same setup for several instruments | Determination of relative lamp position is difficult (sometimes impossible) Finding suitable locations is difficult Rather large effort Only during nighttime | ±0.05° |
| Near lamp | Artificial light source close to the instrument (a few meters) during nighttime | Horizontal distance to lamp (in the order of ±5 cm) Vertical lamp position (around ±0.5 cm, depending on exact setup) | Rather accurate and precise Easy setup Little effort | Close distance to telescope Only during nighttime Only one instrument | ±0.12° |
| Horizon scans | Visible horizon during daytime is used as reference target | Horizontal distance to visible horizon (in the order of ±50 m) Position of visible horizon (around ±5 m) | Easy implementable Applicable regularly Monitoring of calibration | Determination of horizon position is difficult Inhomogeneous horizon hinders exact determination of the horizon Dependence on sky conditions | ±0.25° |
| White stripe | White stripe in front of a black/dark background during daytime | Horizontal distance to stripe (in the order of ±5 cm) Vertical stripe position (around ±5 mm, depending on exact setup) | Accurate and precise (with optimised setup) During daytime | Large effort | ±0.1° |
| Sun scans | Solar disc is used as reference target | Geo location (in the order of 100 m) | Accurate and precise Applicable regularly Elevation + azimuth calibration | Only for 2D instruments Rather complex Only possible for clear sky | ±0.05° |

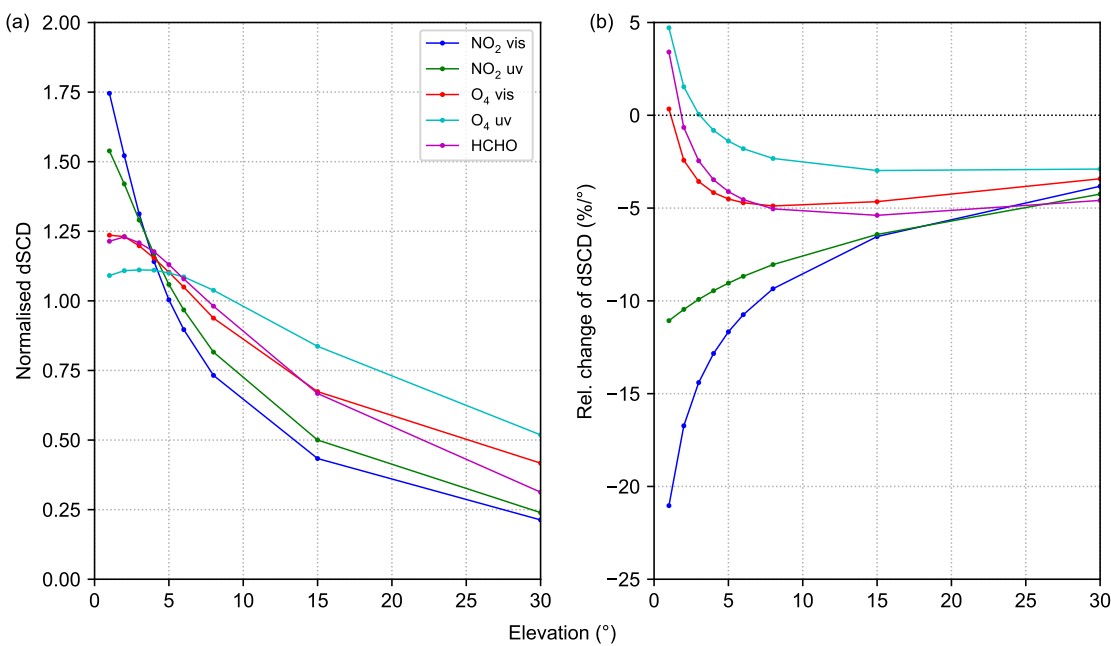

**Figure 1.** (a) Mean dependence of the trace gas dSCDs on the elevation angle for the whole campaign. Each curve was normalised to the mean dSCD value of the elevation sequence. (b) Relative change of the measured dSCDs per degree pointing error at different elevation angles.

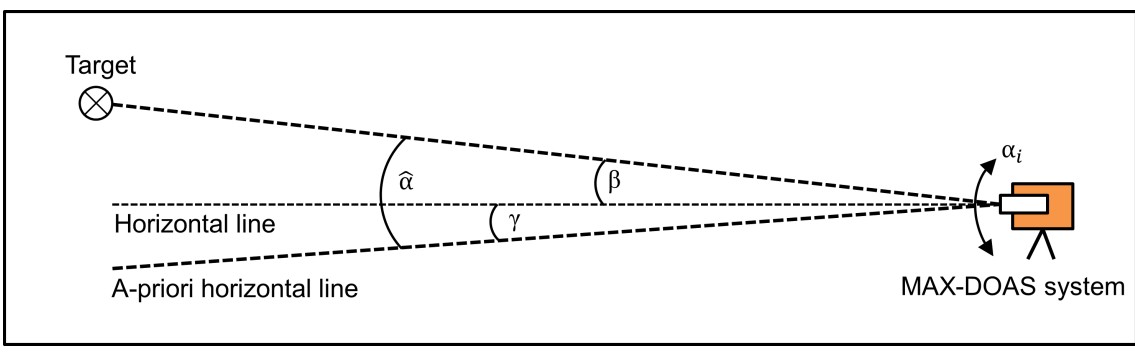

**Figure 2.** Sketch of the general experimental setup and angle definition used in this study.

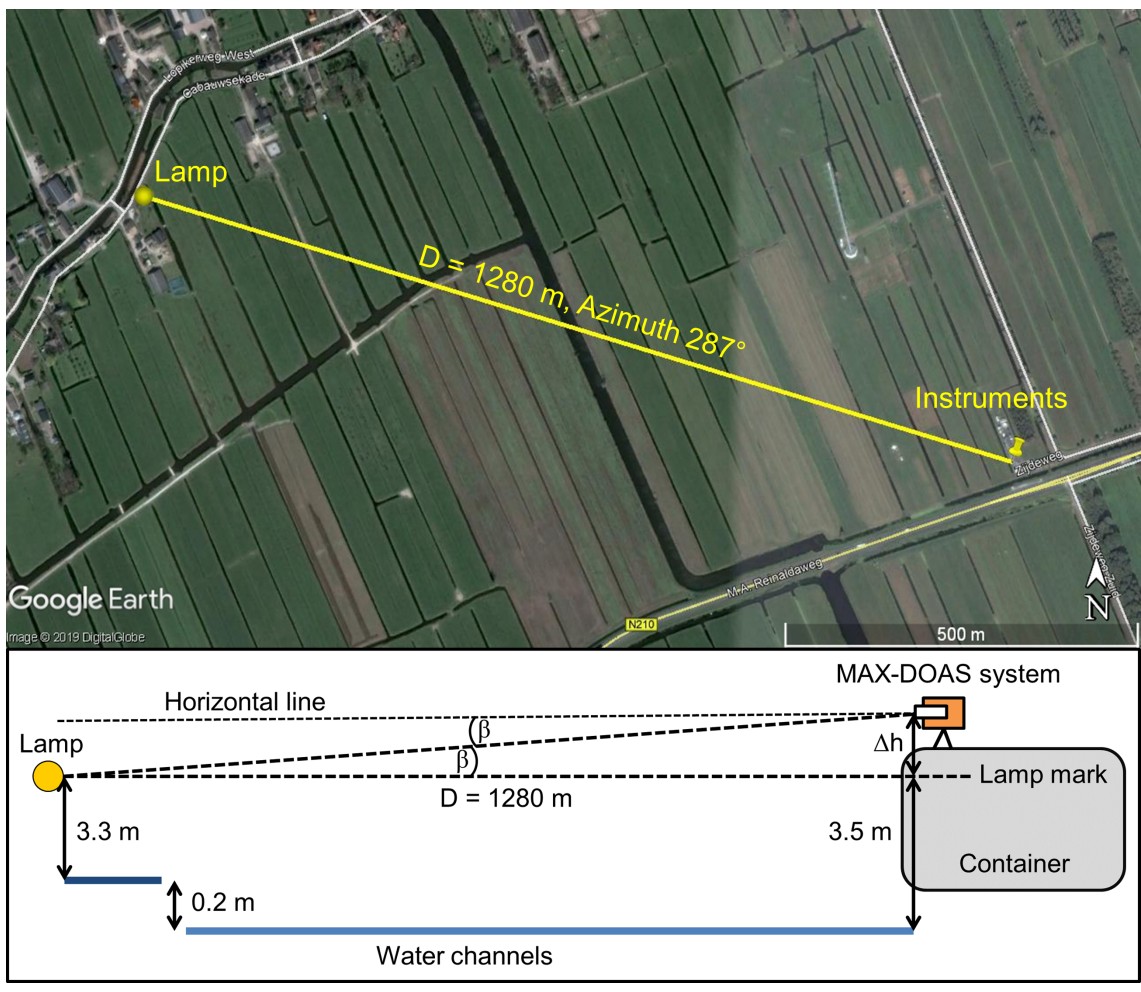

**Figure 3.** Upper panel: map indicating the position of both the lamp and the MAX-DOAS instruments. Lower panel: sketch of the setup for the far lamp measurements during the CINDI-2 campaign. Note, that the lamp is placed below the height of the telescope and therefore a slightly negative elevation of −0.05° is expected (see text).

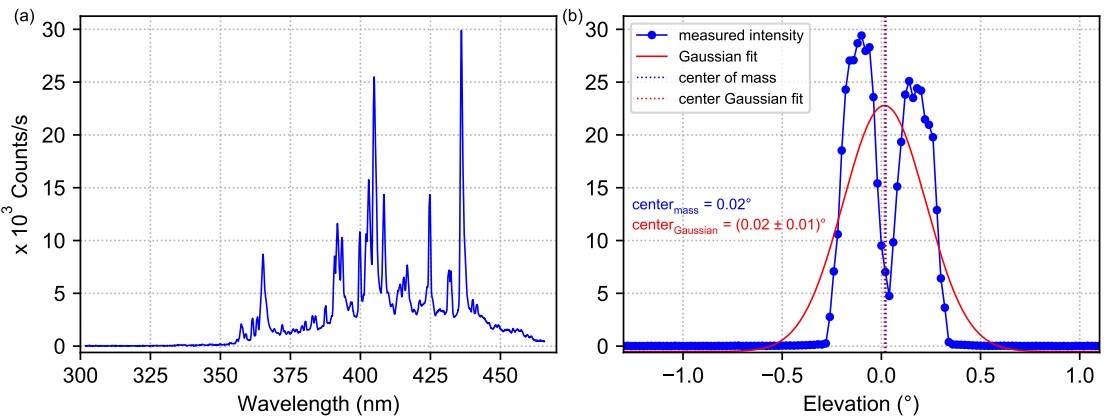

**Figure 4.** (a) Xenon lamp spectrum recorded on 13th September 2016 by the MPIC instrument. The three distinct emission lines at 365.16 nm, 404.90 nm and 435.96 nm which are used for the analysis are clearly visible. (b) Intensity curve at 435.96 nm (blue solid line) recorded on 13th September by the MPIC instrument. The obtained center of mass is indicated by the blue dashed line. Further, the center obtained by a Gaussian fit and the corresponding fit are displayed in red. For the explanation of the specific shape of the intensity curve, see text.

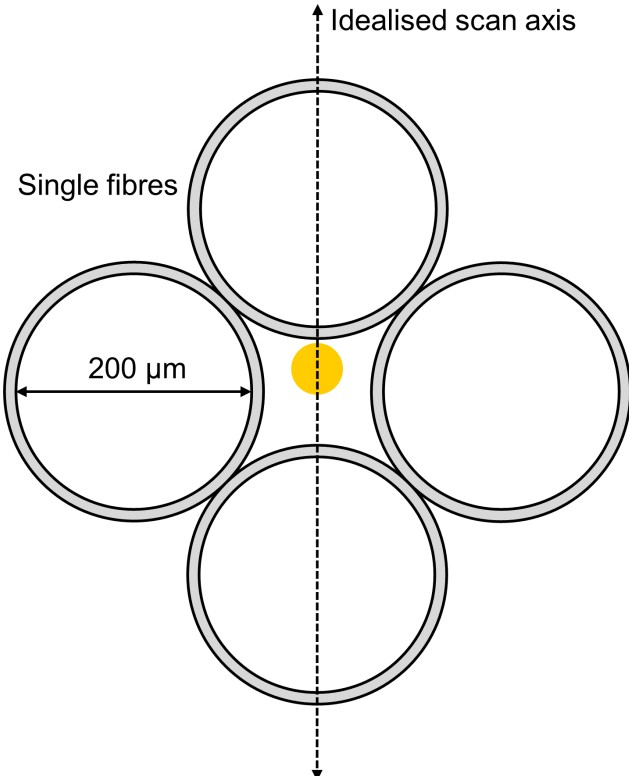

**Figure 5.** Sketch of the fibre bundle placed in the focal point of the telescope of the MPIC instrument. The grey parts indicate the gladding (additional 20 µm) of the fibres. The white circles represent the light-conducting part of the single glass fibres with a diameter of 200 µm, while the yellow spot indicates the idealised image (neglecting aberration etc.) of the xenon lamp inside the telescope which has a diameter of 7 µm. Note, that the size of the yellow dot is not shown at the correct scale relative to the fibre diameter and is larger than in reality.

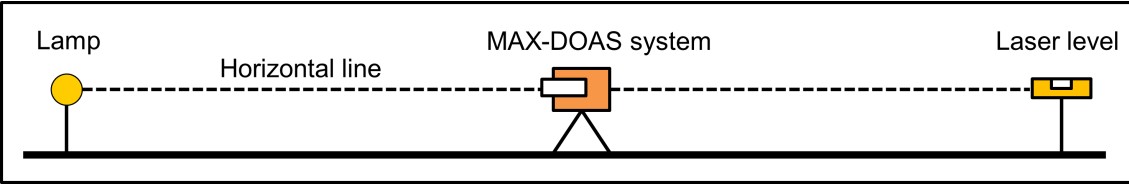

**Figure 6.** Sketch of the measurement setup for the near lamp measurements and the alignment of telescope and lamp.

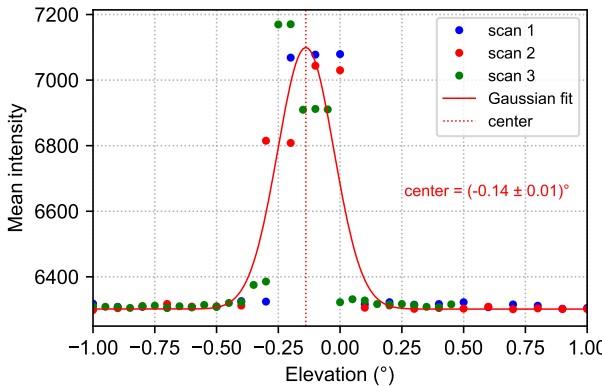

**Figure 7.** Measured intensities for the three individual scans (coloured dots) and the fitted Gaussian (red solid line) obtained from the near lamp measurements by the IUP-HD instrument in the UV spectral range.

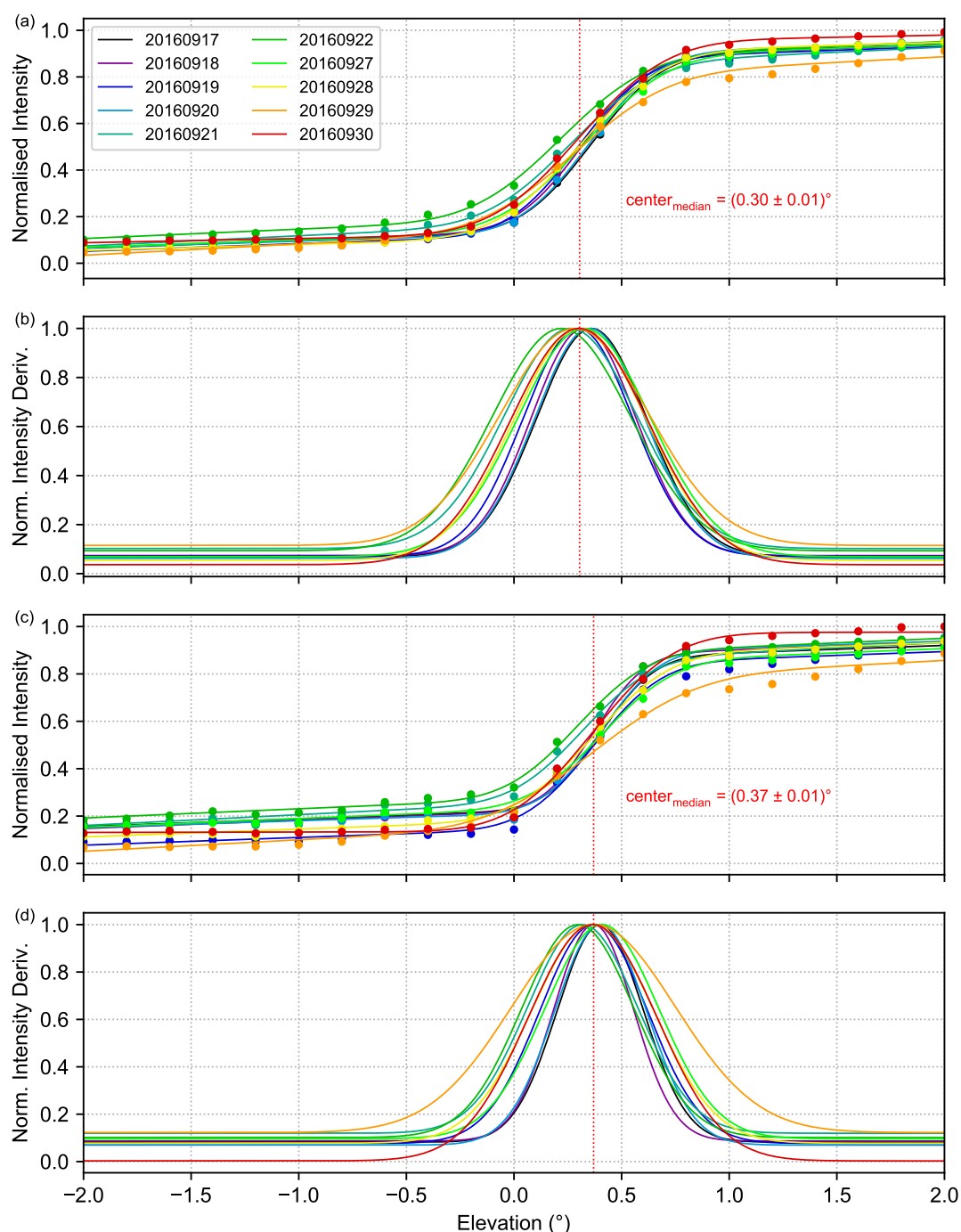

**Figure 8.** (a) and (c) Normalised intensity curves for the horizon scans performed by the MPIC instrument throughout the campaign at 340 nm and 440 nm, respectively. The coloured solid lines indicate the respective Gaussian integral fits. (b) and (d) Normalised derivatives of the respective intensity curves. The median centers of the horizon scans are represented by the red dashed lines.

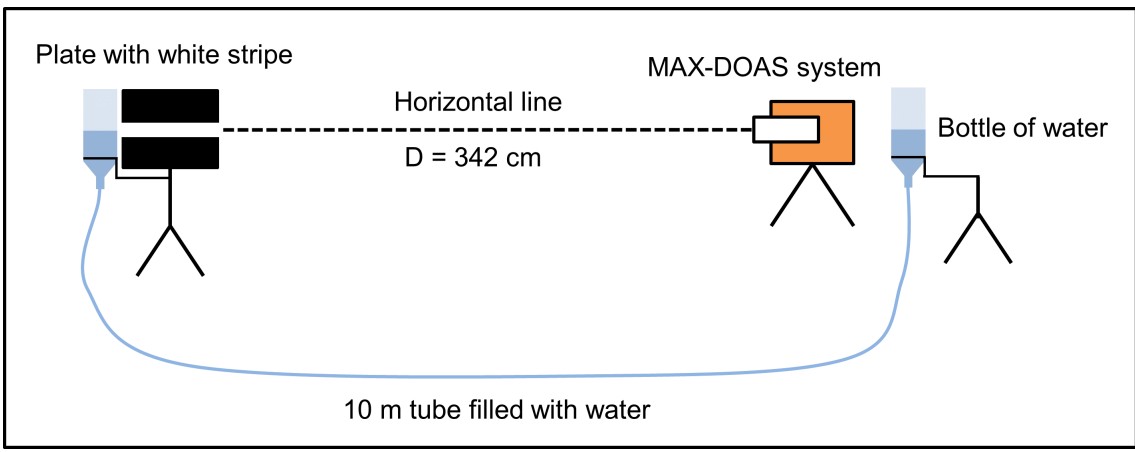

**Figure 9.** Sketch of the measurement setup used by MPIC for the white stripe scans and the alignment of the telescope and white stripe using a water level.

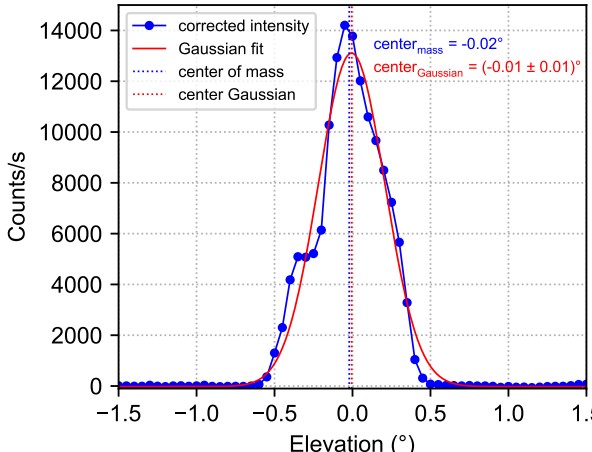

**Figure 10.** Background corrected intensity curve at $440\,\text{nm}$ and corresponding Gaussian fit for the white stripe scan on $20^{\text{th}}$ September performed by the MPIC instrument. The retrieved apparent stripe positions for the two methods are indicated by the dashed lines.

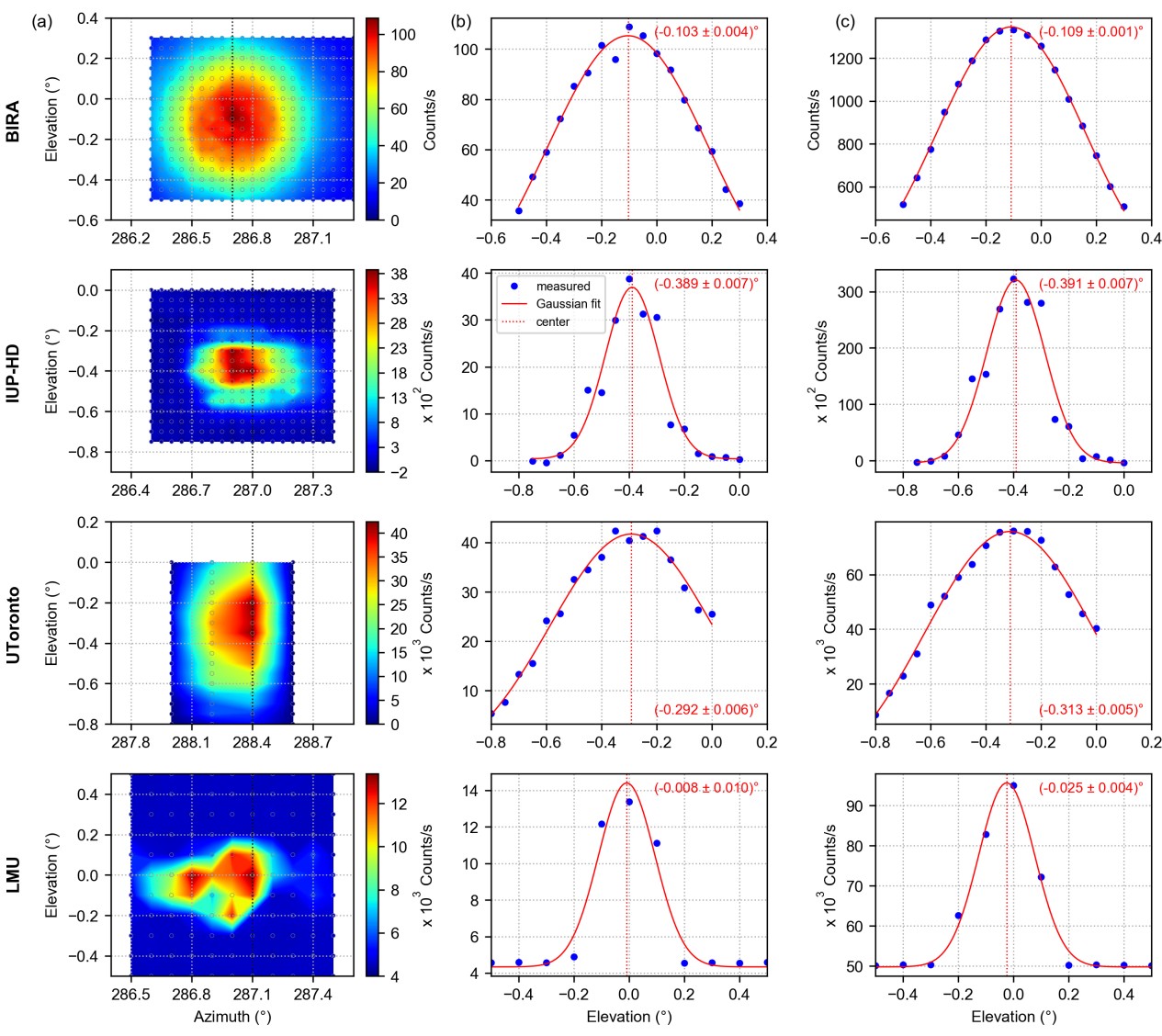

**Figure 11.** Column (a) shows examples of the 2D intensity distributions for the BIRA UV (at 365 nm, measured on 10th September), IUP-HD UV (at 365 nm, measured on 13th September), UToronto (at 436 nm, measured on 10th September) and LMU (mean intensity, measured on 10th September) instruments. Columns (b) and (c) show the corresponding transects along the black dashed lines in column (a) and the azimuthal sum of the intensities at the different elevations, respectively. Additionally, the respective Gaussian fits and their centers are indicated.

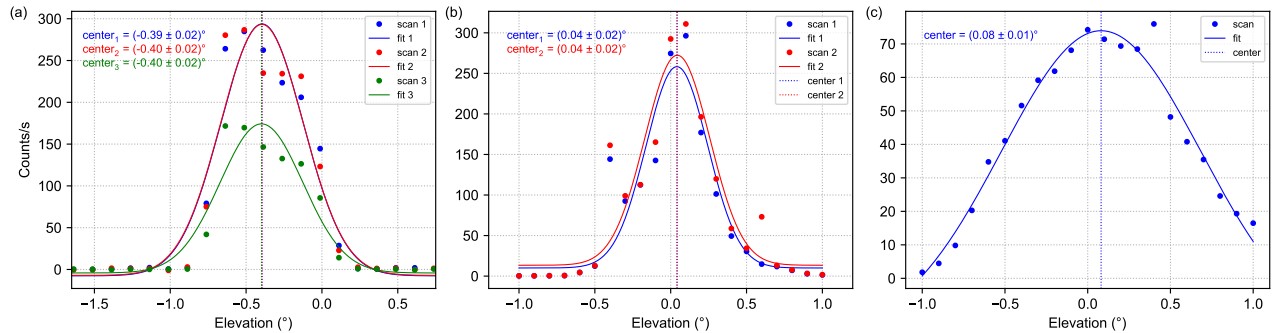

**Figure 12.** (a) Intensity curves at 365 nm recorded on 10th September by the AUTH instrument. (b) Intensity curves at 546 nm recorded on 13th September by the BOKU instrument. (c) Intensity curve at 546 nm recorded on 13th September by the IUP-B visible instrument.

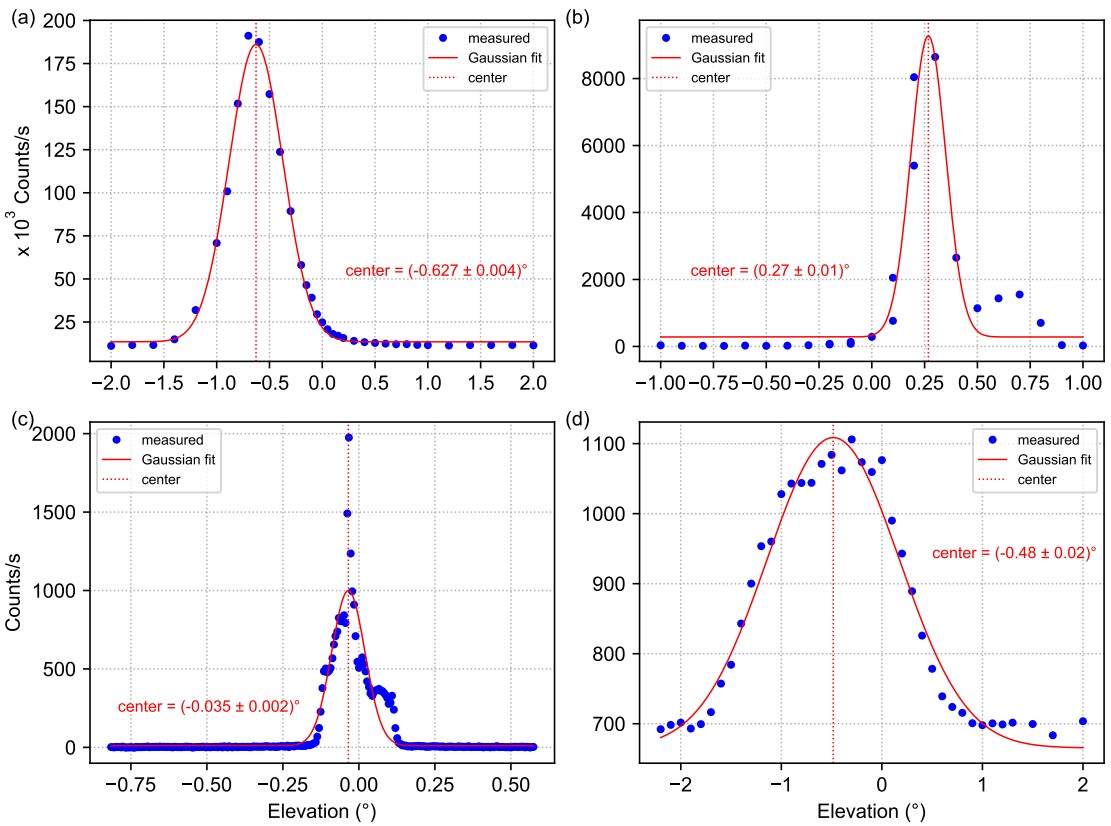

**Figure 13.** (a) Intensity curve at 436 nm recorded on 19[th] September by the CMA UV instrument. (b) Intensity curve at 546 nm recorded on 19[th] September by the CMA VIS instrument. (c) Intensity curve at 365 nm recorded on 10[th] September by the BSU instrument. (d) Intensity curve at 365 nm recorded on 8[th] September by the AIOFM instrument.

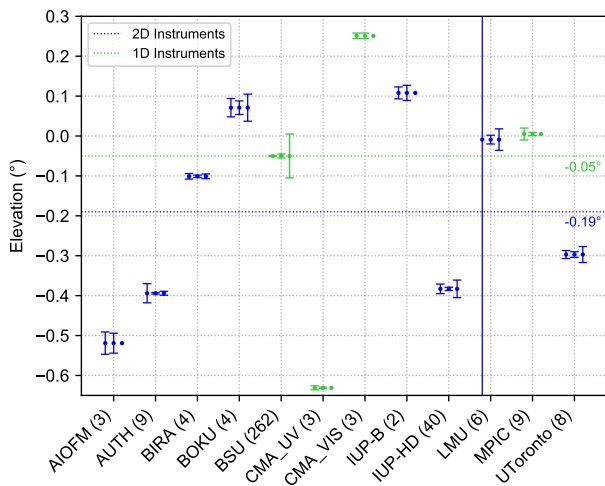

**Figure 14.** Overview of the retrieved lamp elevations for the 2D and 1D instruments including different measures of their uncertainty, mean of fit errors (left), error of the mean (middle) and daily spread (right). The number of available lamp scans for each instrument is displayed in brackets after the individual institute acronyms. The expected lamp elevations are indicated by the corresponding dashed lines.

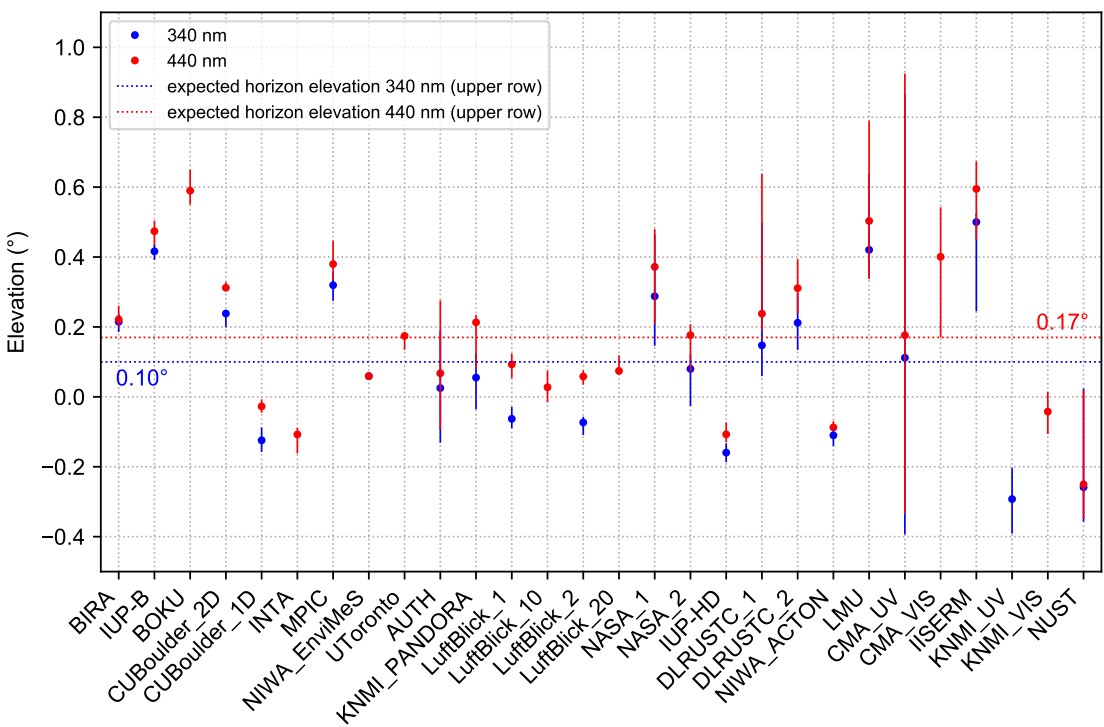

**Figure 15.** Overview of the retrieved median horizon elevations for all participating instruments at the two different wavelengths. Additionally, the 25 % and 75 % percentiles are indicated by the lines.

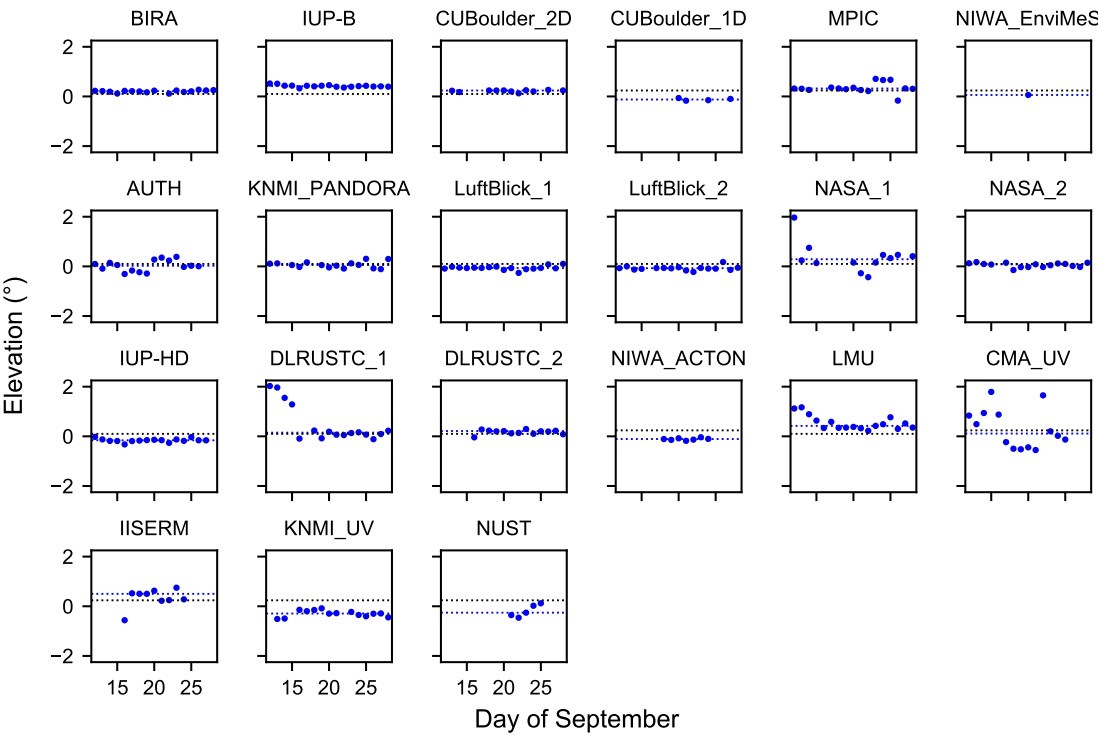

**Figure 16.** Overview of the daily results of the horizon scans for all participating instruments at 340 nm. Additionally, in each subplot the median horizon elevation (blue dashed lines) and expected horizon elevation (black dashed lines) for the corresponding instrument are indicated.

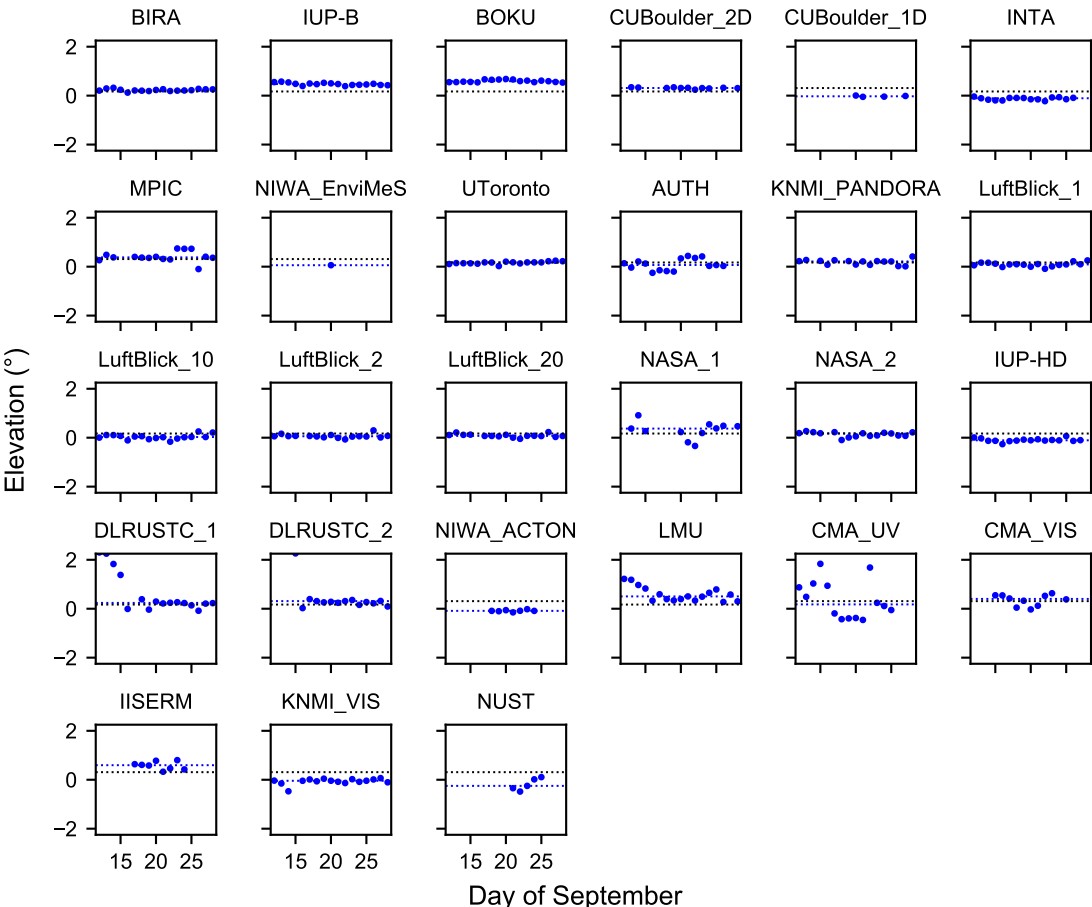

**Figure 17.** Overview of the daily results of the horizon scans for all participating instruments at 440 nm. Additionally, in each subplot the median horizon elevation (blue dashed lines) and expected horizon elevation (black dashed lines) for the corresponding instrument are indicated.

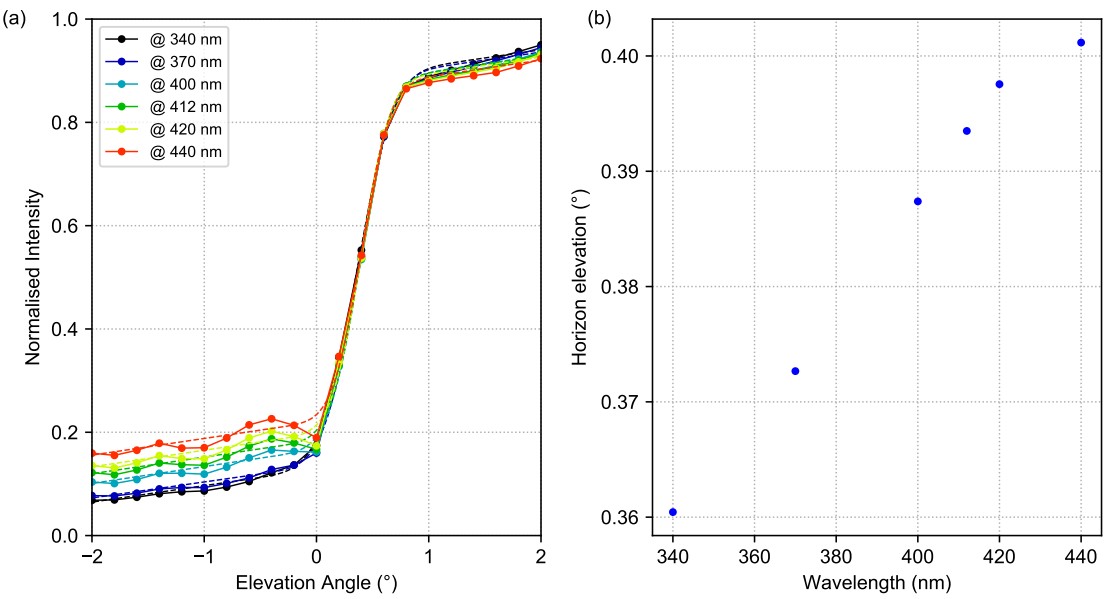

**Figure 18.** (a) Normalised intensity curves (coloured dots) and corresponding Gaussian integral fits (coloured dashed lines) at different wavelengths for the horizon scan performed on 17[th] September by the MPIC instrument. (b) Respective horizon elevations (retrieved from the fits in (a)) as a function of wavelength.

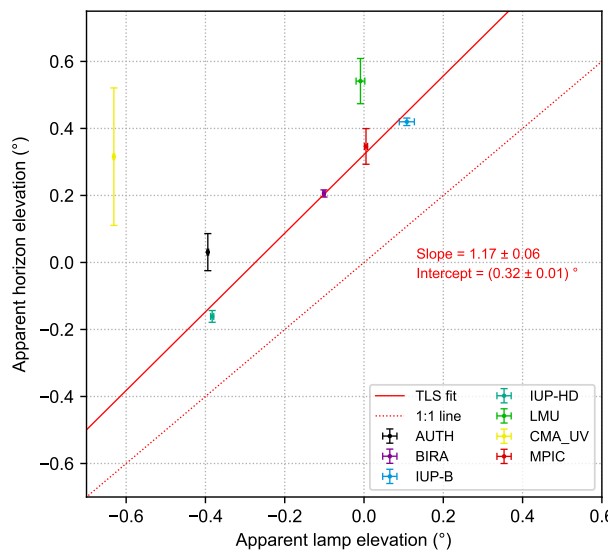

**Figure 19.** Correlation between retrieved horizon (at 340 nm) and lamp elevations for all participating instruments that reported results for both methods. The error bars indicate the standard errors of the mean lamp and horizon elevations. Further, the result of a TLS fit and the 1:1 line are shown.

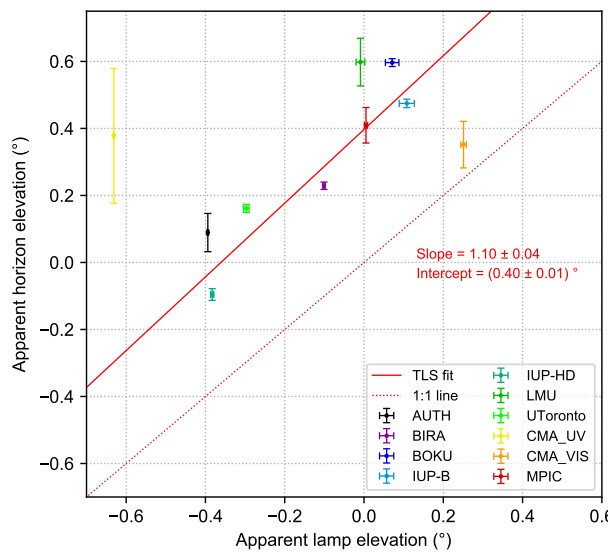

**Figure 20.** Correlation between retrieved horizon (at 440 nm) and lamp elevations for all participating instruments that reported results for both methods. The error bars indicate the standard errors of the mean lamp and horizon elevations. Further, the result of a TLS fit and the 1:1 line are shown.

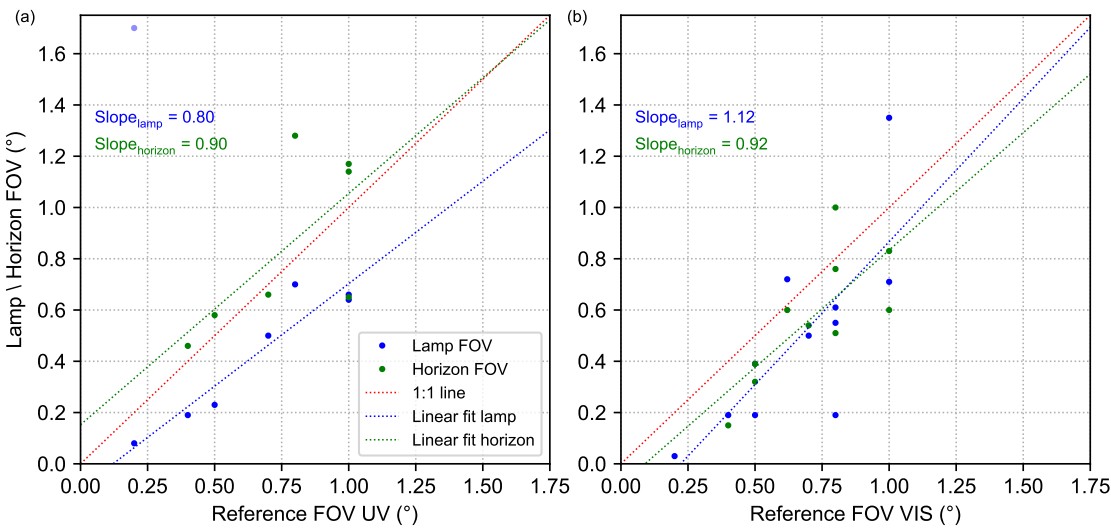

**Figure 21.** Correlation between retrieved FOVs from horizon scans (green dots) and far lamp measurements (blue) against their corresponding reference FOVs reported by the groups. Panel (a) shows the results for the UV spectral range, while panel (b) displays the same for the visible spectral range. In both plots linear regressions and 1:1 lines are included in the respective colours.

**Table A1.** List and evaluation of systematic effects influencing the measurements presented in this paper.

| Effect | Evaluation | Comment |
|---|---|---|
| Uncertainty related to the determination of the target position | Far lamp: ±0.5 m ⇒ ±0.02°<br>Near lamp: ±2 mm and ±0.1° ⇒ ±0.11°<br>Horizon scans: ±5 m ⇒ ±0.22°<br>White stripe: ±5 mm ⇒ ±0.08° | Applies to all methods |
| Motor uncertainties | Approaching positions from different directions (upwards/downwards) can cause large systematic errors (e.g. 0.4° for the MPIC instrument). | Negligible if positions are always approached from the same direction for most instruments. |
| Errors associated with the Gaussian fits (standard errors of the retrieved centers); systematic if intensity curves are asymmetric | Ranging from ±0.01° to roughly ±0.1° depending on the instrument (good an worse ones). | Applies to all methods. However, for many of the instruments it is in the order of ±0.01°. |
| Atmospheric refraction | The effect of atmospheric refraction can become large for almost horizontal light paths. For example, for observations of the sun or the moon deviations of about 0.5° occur for observations close to the horizon. | Not important for white stripe and near lamp measurements. Since for the horizon scans and the far lamp measurements the horizontal distance is much shorter than for the observation of extraterrestrial objects close to the horizon, the effect of refraction is also very small: <0.005°. |
| Visibility | Low visibility influences measurements where the target is located at a far distance from the instrument. | Not important for white stripe and near lamp measurements. If only good conditions are selected, also negligible for horizon scans and far lamp measurements. |

**Table A2.** List and evaluation of statistical effects influencing the measurements presented in this paper.

| Effect | Evaluation | Comment |
| --- | --- | --- |
| Motor uncertainties | Figures 16 and 17 show that for most instruments the daily horizon elevations can be reproduced quite well and the values scatter rather closely around their median value. The reproducibility of the horizon elevations is roughly 0.02° to maximum 0.1° (depending on the instrument performance). | Applies to all methods |
| Statistical uncertainties of the spectral measurements itself + Errors associated with the Gaussian fits (standard errors of the retrieved centers); statistical if intensity curves are very noisy | Should be negligible due to sufficiently long integration times. | Applies to all methods, but can usually be neglected. |
| Atmospheric turbulence | For light paths close to the surface the turbulence elements have dimensions of the order of the distance between the light path and the ground (meter range). For such turbulence elements the temporal fluctuation is of the order of several seconds. | For near lamp and the white stripe measurements atmospheric turbulence can be neglected. For horizon scans and far lamp measurements, turbulence can in principle become important. However, since the length of the light path is orders of magnitude longer that typical turbulence elements, the effects of different turbulence elements along the light path cancel each other out. |