# Peer review of "Evaluating different methods for elevation calibration of MAX-DOAS instruments during the CINDI-2 campaign"

_Atmospheric Measurement Techniques, 2019_

## Referee Comment (RC1) · Anonymous Referee #1 · 6 Aug 2019

This paper by Donner et al. describes and evaluates four different methods for elevation angle calibration of MAX-DOAS instruments in the field. It evaluates the four methods using multiple MAX-DOAS instruments during the CINDI field campaign and concludes that all four methods are suitable for field calibration. Since there is a lack of papers documenting calibration of MAX-DOAS elevation angle, and this paper would be a first step towards standardizing elevation angle calibration for MAX-DOAS instruments, I suggest the paper be accepted with minor changes.

Major comments:

The paper lacks description of laboratory calibration methods used before the field

campaigns. It is mentioned that the instruments are calibrated in the lab before deployment on the field but no descriptions are provided. I think it would be very helpful to have some description of laboratory calibration methods and compare the pros and cons of the laboratory vs field calibration methods. What are the challenges of reproducing laboratory calibration methods? Is laboratory calibration better than field calibration? What is the accuracy and precision of laboratory calibrations?

The paper concludes that all four methods are suitable for field calibration of MAX-DOAS instrument even though the horizon scans could result in an offset of 0.3-0.6 degrees. In the introduction, it is mentioned that a 1 degree error in elevation angle at 0 degree elevation angle could result in $\sim$20% error in NO2 dSCD. Thus, recommending a method which could result in similar bias that the authors are trying to minimize does not seem right. I suggest the authors modify their conclusions to reflect this information provided in the introduction. I also suggest recommending one or two methods based on uncertainty of the method and ease of implementation.

The far lamp, near lamp and white stripe test all relies on knowing the optical axis of the instrument for accurate calibration. If the optical axis of the instrument is well known, could you calibrate the elevation angle using a bubble or digital level? Did any group level their instruments using such a level before the field campaign? I think this is a very important information as bubble or digital level is the simplest way to calibrate elevation angle. So, how does this method compare with the methods presented in the paper?

The paper simply glossed over the backlash issue. This is especially important for the 2D-MAX-DOAS instruments which are capable of doing elevation angle scans at any azimuth angles. The authors found a 0.4 degree difference between scanning from the bottom vs top and decided to just scan from the bottom. I think 0.4 degree is quite significant. How many of the instruments suffer from such a backlash issue? I suggest the authors include some comments/best practices to avoid such issues especially for a 2-D MAX-DOAS?

Minor Comments:

P1, Line 5: This method was applied to more than 12 instruments as can be seen from Figure 19-21. Why did you not include all the instruments in the paper? I suggest make the number of instruments in the paper consistent.

P4, Line 27: How accurate are these inclinometers? I would think the motor steps are more accurate than the inclinometers.

P5, Line 20: What was the resolution used for the initial calibration?

P7, Line 10: How is the light source aligned with the optical axis of the instrument using the laser level?

P7: Near lamp measurements: What are the pros and cons of near lamp measurements? What is the expected accuracy of near lamp measurements? I think the near lamp measurements need to be described in detail as it is likely easier to set up.

P9, Line 5: How does the calculated FOV compare to the reported FOV in Table 2 for different instruments?

Section 4.1: I think the main message of this is lost amongst different type of instruments and different scanning modes. Please consider making this section concise. The main message of this section is (1) independent laboratory calibration between different groups agree within 0.9 degrees, and (2) far lamp calibration method is stable. I don't think all the figures are needed to convey these points.

P14, Line 4-6: It seems that 0.9 degree spread is related to the initial laboratory calibration and not FOV? Why do you think this is related to the FOV?

P14, Line 28 – P15, Line 17: A lot of text to say we don't know what is going on. And it does feel like the author is rambling at times. Please be concise. May be it is better left for a separate paper.

P14, Line 18: Change section title to "comparison between far lamp and horizon elevations" as there is already a comparison between methods section.

P17, Line 13: How do you come up with 0.1 degree uncertainty? Based on the far lamp and near lamp measurements results for IUP-Hd, there is a bias of -0.3 degrees for the near lamp measurements?

P18, Line 2: How do you come up with 0.1 degree uncertainty? Is this an estimate of reproducibility error? What is the uncertainty of the Gaussian fit in figure 11?

Table 1: Please remove instrument ID. While it saves the authors from remaking Figure 21 and 22, it is very confusing to the reader. There are already too many acronyms and having a suffix that is not needed is not helpful.

Table 3: Was the instrument calibrated using the far lamp before 08/09? Are these just test of reproducibility?

Figure 3: Based on the lower part, the elevation angle of the lamp should be negative?

Figure 5: Why are the measured intensity not symmetric? How does this asymmetry affect instrument elevation angle calibration?

Figure 7: How do you make sure the laser level is aligned with the optical axis of the instrument to calibrate the elevation angle?

Figure 9: Please mention what is the red dashed line in the figure?

Figure 11: This shows that it is the best method? Why is this not recommended exclusively?

Figure 12: Panel c: I think it might be better to show average than sum? Same with Figure 13-15.

Figure 12-15: I think it would be better to combine these into one figure. Also, all these figures are likely not needed.

Figure 18: What is mean of fit errors? Instrument label is confusing?

Figure 19: There are instruments that are not listed in Table 1. Make the markers little larger. How is the expected horizon calculated?

Figure 20 and 21: Add the expected horizon on the plot. There are instruments in the plot that are not listed in Table 1. Either add them to Table 1 or remove from the figure.

Figure 24 and 25: Why are the error bars for different instruments so different? It seems like there were different number of measurements for different instruments. I think it would be more appropriate to include standard error of mean as the error bar.

---

## Referee Comment (RC2) · Anonymous Referee #3 · 22 Aug 2019

"Evaluating different methods for elevation calibration of MAX-DOAS instruments during the CINDI-2 campaign" by Sebastian Donner et al. 2019 addresses a very important topic of pointing accuracy of the MAX-DOAS instruments. The authors describe and compare four different methods to measure offset of the actual pointing from the expected pointing. This topic is very important for reliability and consistency of MAX-DOAS observations and is within the scope of the Atmospheric Measurements Technics. I strongly believe this paper can become a great resource for MAX-DOAS community. I recommend publishing it after some changes. Major comments:

1. While the described methods are intended as calibration of instrumental pointing

accuracy no discussion of the instrument performance evaluation after calibration was applied is presented.

2. The paper mainly focuses on determination of the instrument specific apparent elevation angles of the target and inadequately addresses the errors associated with determination of the absolute position of the calibration targets.

3. The authors often use method precision to describe accuracy. To evaluate usefulness of the presented methods both are needed.

4. Measurements of distances, heights, estimation of water levels have no associated with them measurement accuracy and precision reported. Sometimes details how these measurements or estimations were conducted are missing completely.

5. Fits of Gaussian functions to data have no fitting errors reported.

6. Five Pandora instruments (1 KNMI, 2 LuftBlick and 2 NASA) during CINDI-2 were performing sun scans on a regular basis (once per hour) to actively calibrate their azimuth and zenith pointing. This method should also be described for comparison with the other methods.

7. More emphasis should be placed on the quality of the positioners.

8. More explanation needs to be provide on how exactly the horizon scans can be used as a calibration tool (considering dependence on the FOV, scattering conditions, uncertainty in underlying surfaces, light incident angles, and true horizon).

9. Paper can be reorganized to be more concise. Some of the tables and figures can be merged (e.g. Table 1 and 2) and some eliminated at all. Text has some redundancy and needs proofreading. I recommend creating a table with a summary of each method including: (1) setup and "absolute" prerequisites; (2) measurements needed, their typical accuracy and precision, data analysis involved; (3) advantages; (4) disadvantages; (5) overall expected accuracy and precision of zero-elevation calibration based on CINDI-2 data for different types of instruments.

Minor comments:

P2, L31-32: Do any of the instruments have laboratory done FOV scans? It will be interesting to compare field of view between the lab and in the field

P3, L13-14: This sentence is unnecessary

P3, L28-29: Figure 19 has data for 28 instruments. Why Table 1 lists only 12? Tables 1 and 2 should be combined

P3, L31: There is no need to cite the URL when Kreher et al is already cited.

P4, L2: This sentence is redundant

P4, L5: Five Pandoras participated in CINDI-2. Each of them performed sun scans as part of routine operation that served as azimuth and elevation calibration. This method should be also presented for comparison.

P4, L16: What is "horizontal line of the telescope"? Is it the optical axis of the telescope/fiber setup when the instrument points at zero-degree elevation angle? How do you determine it?

P4, L30: This is another reason why sun scanning by Pandora instruments should be discussed in this paper.

P5, L6: How were the distances measured from the lamp to the instruments? How were the vertical distances measured? The land and the canal banks were covered with grass and are not perfectly flat. What is the uncertainty in all distance measurements?

P5, L8: Information from Fig 2 can be communicated in Fig 3 and Fig 2 removed.

P5, L7: The word "compare" next to most Fig and Table references is unnecessary.

P5, L8: The lamp light was collimated and then "directed". How exactly was this achieved? What was the accuracy of the lamp pointing? How uniform was the resulting beam that was visible on the container?

P5, L19: Which earlier night, 8 Sep 2016, or before that? Is 0.16 deg the offset from the initial "a prior" calibration done in the lab or from the earlier night?

P5, L21: Fig 4 is unnecessary and should be removed.

P6, L14: How did you decide that one direction was better than the other? What was positioning error of the lamp? Please include characteristics of instrument positioners (manufacturer accuracy and precision, and used resolution) in Table 2.

P6, L18: Is 0.16deg the initial calibration? Or is this the effect of the positioner resolution?

P7, L9-10: What is the leveling accuracy of the laser level? What light source is used? How is uniformity of the beam achieved? How accurate is determination of the light source center? What are the requirements of light source installation? It is also assumed that the optical axis of the telescope/fiber setup co-align with the mechanical center of the telescope (e.g. the fiber however can be slightly higher or lower than this estimation). What is the final error in determining this betta (= zero) offset angle between the center of the telescope/fiber and the lamp?

P7, L18-19: Were all the scans done from the same direction (upwards or downwards). Looking at the intensities for scan 2 and 3 they might be an indicator of the positioner backlash or pointing issues.

P9, L5: It is not quite clear how this FOV determination eliminates dependencies on the scattering conditions (wavelength), underlying surfaces and their albedo, as well as solar position.

P10, L23. Should "horizon" be replaced with "horizontal"?

P10, L26: How was "visible horizon" determined? How was "closeness" to the visible horizon determined? Figure 3 suggests that the lamp was at 3.3 m above ground. Is

"ground level" referring to 3.3 m above ground?

P10. L29. How was 6.5 to 8.0 m estimated? P11, L13: I recommend combining Fig 12-15 into one figure. These figures give a good sense of the apparent FOV, as the lamp is scanned, impacted by both optics and precision of the positioner. Selecting the azimuth with the maximum intensity is somewhat arbitrary for some of the instruments with asymmetric FOV (e.g. Fig 13).

P11, L20: Differences in positioner pointing precision is also an important parameter of the apparent FOV

P11, L21: Fig 12 suggests that FWHM is about 0.65 deg for the BIRA instrument not 1 deg.

P11, L22: I would say good alignment, center "spot" fiber arrangement and good positioning are the reasons for a relatively uniform FOV. Unless the lamb beam was not uniform.

P12, L23: Isn't the spread expected due to differences in prior reference calibration of 0 elevation angle?

P13, L9: Precision (repeatability) will be more appropriate here instead of "accuracy"

P13, L16: I would replace "measure of consistency" with: "measure of variability"

P13, L22-23: How were 6.5 and 8 m as a function of wavelength (340 and 440 nm) estimated?

P14, L32: Reading the text that follows you assert this difference is due to surface reflectivity. I also will add effects of FOV and wavelength dependent scattering. For a system with a 0.6 deg field of view placed directly on the ground pointing at 0 deg elevation angle (assuming no obstacles): half of the FOV will receive photons scattered in the atmosphere and half reflected from the underlying surfaces. Since Rayleigh scattering is wavelength dependent more photons at longer wavelength would be scattered

in the "above ground" half of FOV. So for the instruments with FOV $\sim$ 0.6 nm FWHM the telescopes should point at least 0.3 deg below horizon to minimize the effect of FOV size. Considering that the instruments during CINDI-2 were located $\sim$ 4 m and 7 m above ground this angle should be even larger and depends on the wavelength and distance to obstacles and solar position (SZA and RAA).

P14, L39: Figure 19 does not support this statement. BIRA_4 instrument horizon position is about the same for both wavelengths.

P15, L25-31: It is unclear how apparent horizon measurements can be suitable for pointing accuracy calibration.

P16, L5: The conclusion maybe applies to the better performing instruments, while the rest of them mostly excluded from the analysis.

P16, L12-13: I do not agree that all the instruments showed consistent results. TLS mainly derived dependence for the better performing instruments... Also the authors have not demonstrated that the instruments improved their pointing performance as a result of any of these calibration methods.

P16, L33: I would not call this method accurate since some of the instruments clearly showed asymmetric FOV and different functions other than Gaussian could describe the intensity distribution potentially leading to larger errors.

P17, L1: Precision should not be confused with accuracy. In my opinion the authors have not accounted for all the uncertainties to claim accuracy of +-0.05 deg.

I recommend combining Table 1 and 2 and adding 2 columns with positioner maker, accuracy and precision data.

I do not think Tables 3 and 4 are needed.

Table 5: I think replacing "row" with "container level" might be clearer.

Figure 2 and 3 should be combined

Figure 3: Upper and lower "panels" instead of "parts"

Figure 4 is unnecessary

Figure 10: It might be good to raise MAX-DOAS instrument in line with the stripe

Figures 12, 13, 14, 15 should be combined to give better idea about apparent FOV for different instruments

Figure 16: intensity distributions in (a, b) clearly do not have Gaussian distributions, please add fitting errors.

Figures 24 and 25: Y-axis = Apparent horizon elevation; X-axis = apparent lamp elevation. Add fitting errors to slope and intercept.

---

## Author Comment (AC1) · 28 Nov 2019

**Reply to comments from Referee #1**

*This paper by Donner et al. describes and evaluates four different methods for elevation angle calibration of MAX-DOAS instruments in the field. It evaluates the four methods using multiple MAX-DOAS instruments during the CINDI field campaign and concludes that all four methods are suitable for field calibration. Since there is a lack of papers documenting calibration of MAX-DOAS elevation angle, and this paper would be a first step towards standardizing elevation angle calibration for MAX-DOAS instruments, I suggest the paper be accepted with minor changes.*

We would like to thank the referee for this positive assessment and several helpful comments.

**Major comments:**

*The paper lacks description of laboratory calibration methods used before the field campaigns. It is mentioned that the instruments are calibrated in the lab before deployment on the field but no descriptions are provided. I think it would be very helpful to have some description of laboratory calibration methods and compare the pros and cons of the laboratory vs field calibration methods. What are the challenges of reproducing laboratory calibration methods? Is laboratory calibration better than field calibration? What is the accuracy and precision of laboratory calibrations?*

Thanks for this helpful comment. We have asked all groups how they calibrated their instruments in the laboratory. 7 groups responded and a short overview on their results was added to section 3.1 "General approach". Additionally, the following statement (P.2 L.32-P.3 L.2) in the introduction part of the paper:

"In the past, these calibrations were mainly done in laboratories where fixed target points were used as references and the elevations were calibrated accordingly. However, when the instruments were brought to the field, only rarely (if at all) the accuracy of the a-priori elevation angle calibration was checked under real measurement conditions."

was changed to:

"In principle, these calibrations can be best done in the laboratory under stable and controlled conditions, where fixed target points are used as references and the corresponding elevations can be calibrated accordingly. In particular, the FOV should be determined already in the laboratory. Nevertheless, elevation calibration in the field is indispensable, because during transport from the laboratory to the field and during installation on the measurement site, it is likely that the instrument characteristics might change. In the past, however, when the instruments were brought to the field, only rarely (if at all) the accuracy of the a-priori elevation angle calibration was checked under real measurement conditions.

*The paper concludes that all four methods are suitable for field calibration of MAXDOAS instrument even though the horizon scans could result in an offset of 0.3-0.6 degrees. In the introduction, it is mentioned that a 1 degree error in elevation angle at 0 degree elevation angle could result in ~20% error in NO2 dSCD. Thus, recommending a method which could result in similar bias that the authors are trying to minimize does not seem right. I suggest the authors modify their conclusions to reflect this information provided in the introduction. I also suggest recommending one or two methods based on uncertainty of the method and ease of implementation.*

Many thanks for this hint which indicates that our wording was not clear enough. The large spread (0.9°) of the results of the horizon scans is related to differences and errors of the initial laboratory

calibration (or changes during transport to the field). Therefore, they don't represent the accuracy of the horizon scan method. Note that the spread found for the horizon scans is consistent with the one obtained from the far lamp measurements. Moreover, the systematic difference (0.3° - 0.6°) between the horizon scans and the far lamp scans is easily explainable by the height difference of the lamp and the visible horizon. In summary, we conclude that also the horizon scan method is in principle quite accurate as long as favorable measurements conditions, like high visibility and non-rapidly changing clouds are selected.
We made these points clearer in the paper, especially in the conclusions.

*The far lamp, near lamp and white stripe test all relies on knowing the optical axis of the instrument for accurate calibration. If the optical axis of the instrument is well known, could you calibrate the elevation angle using a bubble or digital level? Did any group level their instruments using such a level before the field campaign? I think this is a very important information as bubble or digital level is the simplest way to calibrate elevation angle. So, how does this method compare with the methods presented in the paper?*

The reviewer is completely right that, if the optical axis is precisely known, a water level would be completely sufficient to calibrate the elevation angles. But this assumption is not always fulfilled, e.g. because the fibre bundle is not perfectly centered etc. Because of these uncertainties the in-field calibration of the elevation angles as described in this paper are indispensable. This information was added and more stressed in the "General approach" section (3.1) and conclusions section of the paper.

*The paper simply glossed over the backlash issue. This is especially important for the 2D-MAX-DOAS instruments which are capable of doing elevation angle scans at any azimuth angles. The authors found a 0.4 degree difference between scanning from the bottom vs top and decided to just scan from the bottom. I think 0.4 degree is quite significant. How many of the instruments suffer from such a backlash issue? I suggest the authors include some comments/best practices to avoid such issues especially for a 2-D MAX-DOAS?*

The reviewer is right when stating that 0.4° difference between scanning from below vs. from above is significant for the elevation of the telescope. For that reason we mentioned this issue in the paper. Nevertheless, we have asked all groups for their handling of this issue. The outcome was that depending on the kind of stepper/motor not all instruments suffered from such backlash issues, others actively corrected for this using inclinometers (e.g. LMU) or active sun trackers (e.g. BIRA). Most of the instruments which experienced backlash issues solved this issue by simply scanning always from the same direction (in elevation and azimuth direction) as mentioned in the paper. The effect of backlash (maximum difference between scanning directions) ranges from fractions of a degree to roughly 1°. While the effect is very important for the elevation pointing of the instruments, the effect has a smaller resulting influence on the measurements in the azimuth direction. The information provided here was also added to the paper in section 3.2.2.

**Minor Comments:**

*P1, Line 5: This method was applied to more than 12 instruments as can be seen from Figure 19 - 21. Why did you not include all the instruments in the paper? I suggest make the number of instruments in the paper consistent.*

Thanks for pointing this lack of clarity out. The horizon scans were performed by all instruments during the campaign which followed the standardised measurement protocol and reported them to the referee of the campaign (28 instruments). However, only 12 instruments (from 11 groups)

participated in the far lamp measurements. We removed the number of instruments from the abstract (P.1 L.5). Further, this information was added to the paper in section 2.2.

*P4, Line 27: How accurate are these inclinometers? I would think the motor steps are more accurate than the inclinometers.*

According to the instrument's manufacturer the accuracy of the inclinometers used for the EnviMes instruments is 0.1° and the precision is 0.03°. Depending on the actual properties of the motors the inclinometers are at least as accurate as the motors if not better.

*P5, Line 20: What was the resolution used for the initial calibration?*

Thanks for pointing this lack of clarity out. The pre-calibration was done using a water level during the setup of the instrument. Then the finer adjustment was done using the results of the far lamp scans from 7$^{th}$ (in this night the lamp measurements were tested by our group with an scan resolution of 0.1° but the scanning was done manually), 8$^{th}$ and 10$^{th}$ September. The other two nights can then serve as tests of reproducibility. All values in this paper are given relative to the elevation calibration which was obtained by these finer adjustments and which was finally used for the campaign. This information was added to the text.

*P7, Line 10: How is the light source aligned with the optical axis of the instrument using the laser level?*

As sketched in Figure 7, the laser level illuminates both the instrument and the position of the lamp. The telescope and the lamp are centered around the position of the laser beam. The associated uncertainties of the relative vertical positions between instrument and lamp are estimated to about 0.1° (leveling accuracy of the laser level) and roughly 0.04° (thickness of the laser beam). This information was added to the revised text.

*P7: Near lamp measurements: What are the pros and cons of near lamp measurements? What is the expected accuracy of near lamp measurements? I think the near lamp measurements need to be described in detail as it is likely easier to set up.*

Thanks for this hint. We have revised the description of the near lamp measurements and added some more details and an improved error assessment (see also previous comment). Pros and cons are discussed in the conclusions section of the paper. Further, also the expected accuracy is given there.

*P9, Line 5: How does the calculated FOV compare to the reported FOV in Table 2 for different instruments?*

Thanks for this interesting question. Despite the fact that the determination of the FOV is not the main aim of the paper we added a new section "4.4 FOV determination" and a new figure (Fig. 21) to the paper, where a comparison between the retrieved FOVs (from horizon and lamp scans) and the reference FOV is provided. In general reasonable agreement is found. However, systematically slightly larger FOVs are found for the horizon scans than for the lamp scans. This can - at least for the 1D instruments - be explained by the fact that the far lamp was not always in the center of the azimuth dimension of the FOV. Also, the determination of the FOV seems to be less stable as compared to the determination of the target positions.

*Section 4.1: I think the main message of this is lost amongst different type of instruments and different scanning modes. Please consider making this section concise. The main message of this*

*section is (1) independent laboratory calibration between different groups agree within 0.9 degrees, and (2) far lamp calibration method is stable. I don't think all the figures are needed to convey these points.*

We agree that this section has a lot of figures and descriptions associated with it. However, most of the figures are needed to illustrate the variety of different instruments and scanning schemes. In order to make this section clearer, we have combined Figures 12-15 and added subsections to give more structure to this section.

*P14, Line 4-6: It seems that 0.9 degree spread is related to the initial laboratory calibration and not FOV? Why do you think this is related to the FOV?*

Thanks for pointing this lack of clarity out. This is a misunderstanding. We did not mean that there is a relationship between the deviation of the elevation calibration and the FOV. We just wanted to express that the spread is of the same magnitude as a typical FOV and does actually matter. To avoid this misunderstanding, we removed this statement from the text.

*P14, Line 28 – P15, Line 17: A lot of text to say we don't know what is going on. And it does feel like the author is rambling at times. Please be concise. May be it is better left for a separate paper.*

We agree that this part is quite detailed and might be not completely suited for this paper. However, we want to keep the main message of this part. Therefore, we have removed Figure 23 from the paper. Further, the text was shortened in the revised version of the paper.

*P15, Line 18: Change section title to "comparison between far lamp and horizon elevations" as there is already a comparison between methods section.*

Thanks for this hint. We changed the section title accordingly.

*P17, Line 13: How do you come up with 0.1 degree uncertainty? Based on the far lamp and near lamp measurements results for IUP-Hd, there is a bias of -0.3 degrees for the near lamp measurements?*

The uncertainty of +-0.1° was estimated as described in the paper and is dominated by the errors introduced by the determination of the lamp position as described in section 3.2 and in the conclusions. This value is an estimate for the uncertainty of this method and not a difference/bias to another method. The near lamp scans in the UV and the VIS spectral range agree within 0.03°. The difference between the far lamp measurements (retrieved lamp position is roughly -0.4°) and the near lamp measurements is roughly 0.3° as mentioned by the reviewer. This, however, fits to the expectations, since the IUP-HD instrument should see the lamp below 0° at roughly. -0.2° as described in the paper.

*P18, Line 2: How do you come up with 0.1 degree uncertainty? Is this an estimate of reproducibility error? What is the uncertainty of the Gaussian fit in figure 11?*

As for the previous comment, the uncertainty of +-0.1° was estimated as described in the paper and is dominated by the errors introduced by the determination of the stripe position as described in section 3.5 and in the conclusions. This value was obtained by combining the errors of the Gaussian fit and the error of the determination of the stripe position. It is an estimate for the systematic uncertainty of this method. The reproducibility error, however, is a statistical error which is dominated by the motor precision as described in the new Table A2 and the conclusion section of the paper. Fit errors were added to all figures showing Gaussian fits.

*Table 1: Please remove instrument ID. While it saves the authors from remaking Figure 21 and 22, it is very confusing to the reader. There are already too many acronyms and having a suffix that is not needed is not helpful.*

We have removed the instrument ID from the table. Further, Tables 1 and 2 were merged to a new table 1. Additionally, the corresponding Figures were adjusted.

*Table 3: Was the instrument calibrated using the far lamp before 08/09? Are these just test of reproducibility?*

See answer above for minor comment P5, Line 20. All values in this Table are given relative to the elevation calibration which was finally used for the campaign. This information was added to the text in section 3.2.2.

*Figure 3: Based on the lower part, the elevation angle of the lamp should be negative?*

Yes, this is correct. The estimation of the lamp position (lamp elevation) relative to the instruments is mentioned and explained in the text serval times, e.g. P.6 L.22-30 and P.13 L.3-5. We added a hint to the caption of Figure 3.

*Figure 5: Why are the measured intensity not symmetric? How does this asymmetry affect instrument elevation angle calibration?*

If the fibre bundle would be exactly located in the focus of the lens and under ideal conditions, the image of the lamp would exactly pass through the center of the space between the fibres. However, all this conditions might be not completely fulfilled which leads to a more asymmetric intensity distribution. This is also mentioned in the paper P.5 L.32 – P.6. L7. We added a hint to the caption of the Figure.

*Figure 7: How do you make sure the laser level is aligned with the optical axis of the instrument to calibrate the elevation angle?*

See answer to the previous comments regarding the near lamp measurements.

*Figure 9: Please mention what is the red dashed line in the figure?*

Thanks for this hint. The red dashed line indicates the median center of the horizon scans. This information is given in text. To make this clearer we also added this information to the caption of the Figure.

*Figure 11: This shows that it is the best method? Why is this not recommended exclusively?*

Indeed, this figure suggests that this method is the best, since the presented fit is quite smooth. Nevertheless, the used setup was not optimal and especially the distance between stripe and telescope should be larger. Problems with the used setup are discussed in section 3.5.1. Further, pros and cons are discussed in the conclusion section of the paper concluding that the white stripe scans are a well-suited method if the setup is stable and optimal.

*Figure 12: Panel c: I think it might be better to show average than sum? Same with Figure 13-15.*

Since an average is just a normalised sum and the Gaussian fit will not be influenced by an additional normalisation, we decided to keep the sums in the plot. Therefore, we did not modify the figure.

*Figure 12-15: I think it would be better to combine these into one figure. Also, all these figures are likely not needed.*

Thanks for this suggestion. However, we believe that all these figures are important to give a good overview on the different characteristics of the different instruments and their FOVs, we decided to keep the figures. However, we combined them into one figure.

*Figure 18: What is mean of fit errors? Instrument label is confusing?*

The „mean of the fit errors" is the mean of all fit errors of the Gaussian fits which were applied to the intensity curves as explained in the text (section 4.1., especially P.12. L.28-30). As explained there, the fit error measures the quality of the Gaussian fits and the shape of the measured intensity curves.
Regarding the instrument labelling: the labels are according to Table 1. The numbers in brackets give the number of available lamp scans as indicated in the caption of the figure.

*Figure 19: There are instruments that are not listed in Table 1. Make the markers little larger. How is the expected horizon calculated?*

As already explained above (first minor comment), in total 28 instruments reported horizon scan data. However, Table 1 only lists instruments which explicitly participated in the far lamp measurements.
The marker size was increased. Additionally, the instrument labels were adjusted according to Table 1 (where instrument IDs were removed).
The estimation of the expected horizon elevations is explained on P.13. L.23-24. and P.10 L.27-30. We slightly revised the explanation of the estimation on P.13 and added more details.

*Figure 20 and 21: Add the expected horizon on the plot. There are instruments in the plot that are not listed in Table 1. Either add them to Table 1 or remove from the figure.*

Regarding the number of instruments which performed horizon scans, please see our previous answers regarding this topic. The expected horizon elevations were added to the revised versions of figures 20 and 21. Additionally, the instrument labels were adjusted according to Table 1 (where instrument IDs were removed).

*Figure 24 and 25: Why are the error bars for different instruments so different? It seems like there were different number of measurements for different instruments. I think it would be more appropriate to include standard error of mean as the error bar.*

The reviewer is right, since the error bars are indicating the standard deviations, the number of measurements plays a role here. However, also the performances of the individual instruments determine the actual sizes of the error bars. Nevertheless, we agree to your comment and replaced the standard deviations with the standard error to represent the error bars. The intercepts and slopes are almost unaffected as well as the relative sizes of the errors bars amongst each other for most of the instruments.

---

## Author Comment (AC2) · 28 Nov 2019

**Reply to comments from Referee #3**

*"Evaluating different methods for elevation calibration of MAX-DOAS instruments during the CINDI-2 campaign" by Sebastian Donner et al. 2019 addresses a very important topic of pointing accuracy of the MAX-DOAS instruments. The authors describe and compare four different methods to measure offset of the actual pointing from the expected pointing. This topic is very important for reliability and consistency of MAX-DOAS observations and is within the scope of the Atmospheric Measurements Technics. I strongly believe this paper can become a great resource for MAX-DOAS community. I recommend publishing it after some changes.*

We would like to thank the referee for this positive assessment and several useful comments.

**Major comments:**

*1. While the described methods are intended as calibration of instrumental pointing accuracy no discussion of the instrument performance evaluation after calibration was applied is presented.*

Many thanks for this valuable comment. The instrument performance was assessed in the overview paper by Kreher et al. (2019). Our study focuses on the details of the elevation calibration procedure. This information was added to the paper. Further, a statement was added in the conclusions that in upcoming campaigns strong emphasis should be put on the monitoring and possible correction of the elevation calibration.

*2. The paper mainly focuses on determination of the instrument specific apparent elevation angles of the target and inadequately addresses the errors associated with determination of the absolute position of the calibration targets.*

Thanks for this good comment. The estimates of the errors of the absolute positons of the targets and the corresponding uncertainties of the elevation angles are given as follows:

-White stripe: +/- 5 mm => +/- 0.08 °

-near lamp: +/- 2 mm and 0.1° (provided by IUP-HD) => +/- 0.11 °

-far lamp: +/- 50 cm => +/- 0.02 °

-horizon: +/- 5 m => +/- 0.22 °

The uncertainties of the angles were calculated from the uncertainties using geometry. We added these accuracy estimates for the positions of the targets to the revised version of the paper.

*3. The authors often use method precision to describe accuracy. To evaluate usefulness of the presented methods both are needed.*

An appendix was added mainly consisting of two tables describing and evaluating different systematic and statistical error sources. The main findings are:
- The systematic uncertainties are dominated by the errors introduced by the determination of the target positions and the fit errors
- The statistical errors are dominated by the precision of the used motors/steppers

Based on these findings, different passages in the paper were modified and new ones were added to address this comment. Further, the error assumptions were motivated more clearly.

*4. Measurements of distances, heights, estimation of water levels have no associated with them measurement accuracy and precision reported. Sometimes details how these measurements or estimations were conducted are missing completely.*

The uncertainties of the measurements of horizontal distances are not critical and can be neglected since the absolute horizontal distances are much larger than the vertical distances. However, error estimates were added to the revised version of the paper (see answer to major comment 2). Further, the descriptions of the determination of vertical (and partly horizontal) distances were revised.

*5. Fits of Gaussian functions to data have no fitting errors reported.*

(Standard) fit errors were added to all plots showing Gaussian fits. Additionally, typical values for the fit errors were added to the revised version of the paper, especially in tables A1 and A2 which summarise the error sources.

*6. Five Pandora instruments (1 KNMI, 2 LuftBlick and 2 NASA) during CINDI-2 were performing sun scans on a regular basis (once per hour) to actively calibrate their azimuth and zenith pointing. This method should also be described for comparison with the other methods.*

Thanks for this valuable and important hint. A new section describing the sun scans was added to the revised version of the paper. Further, this method is discussed in the conclusion section of the paper and mentioned at different sections of the revised paper.

*7. More emphasis should be placed on the quality of the positioners.*

Thanks for this interesting comment. However, Figures 20 and 21 show that for most (better performing) instruments the daily horizon elevations can be reproduced quite well and the values scatter rather closely around their median value. The reproducibility of the horizon elevations is typically significantly better than +-0.1°. Given the uncertainties in the determination of the visible horizon this is rather small. The same is found for the far lamp measurements (Figure 18). Here, small error bars are found for the standard error of the mean lamp elevation which also indicates good reproducibility. Therefore, we conclude that the quality of the positioners does not dominate the overall uncertainties and does not put doubt on the findings of this study for most of the instruments. However, we added detailed information on the statistical and systematic error sources to the paper (especially in the two new tables A1 and A2 in the Appendix) for the different methods as described in this answer to the revised version of the paper.

*8. More explanation needs to be provide on how exactly the horizon scans can be used as a calibration tool (considering dependence on the FOV, scattering conditions, uncertainty in underlying surfaces, light incident angles, and true horizon).*

The description of the horizon scans is already quite detailed and also precautions and prerequisites are discussed at several passages of the paper (e.g. P.8 L.8-12 or P.14 L.26-27). However, we agree that this information is quite scattered over the whole paper. Therefore, we added and stressed some precautions and prerequisites which have to be fulfilled to use the horizon scans as a useful calibration tool. Namely:
- Days with good visibility should be used
- Days with rapidly varying cloud cover and/or low-lying clouds should be avoided

*- Another interesting aspect is that with increasing FOV the slope of the horizon scans becomes weaker. However, in spite of this weaker contrast this is in general no problem.*

*9. Paper can be reorganized to be more concise. Some of the tables and figures can be merged (e.g. Table 1 and 2) and some eliminated at all. Text has some redundancy and needs proofreading. I recommend creating a table with a summary of each method including: (1) setup and "absolute" prerequisites; (2) measurements needed, their typical accuracy and precision, data analysis involved; (3) advantages; (4) disadvantages; (5) overall expected accuracy and precision of zero-elevation calibration based on CINDI-2 data for different types of instruments.*

Tables 1 and 2 were merged as well as Figures 4 and 5. Further, Figures 12 to 15 were combined. Figure 23 was removed from the paper and the respective paragraph was shortened in the manuscript. Regarding the proofreading, we agree that both grammar and spelling is not perfect since we are no native speakers, however, Copernicus will provide a proofreading procedure during typesetting. An overview table (new table 5) summarising the different methods, listing advantages and disadvantages and giving uncertainty estimates was added.

**Minor comments:**

*P2, L31-32: Do any of the instruments have laboratory done FOV scans? It will be interesting to compare field of view between the lab and in the field.*

Thanks for this interesting question. The values for the FOVs listed in Table 2 were provided by the groups and were determined in the laboratory. Despite the fact that the determination of the FOV is not the main aim of the paper, we added a new section "FOV determination" to the paper, where a comparison between the retrieved FOVs (from horizon and lamp scans) and the reference FOV is provided. In general reasonable agreement is found. However, the determination of the FOV seems to be less stable as compared to the determination of the target positions.

*P3, L13-14: This sentence is unnecessary.*

The sentence was removed from the paper.

*P3, L28-29: Figure 19 has data for 28 instruments. Why Table 1 lists only 12? Tables 1 and 2 should be combined.*

Thanks for pointing this lack of clarity out. The horizon scans were performed by all instruments during the campaign which followed the standardised measurement protocol and reported them to the referee of the campaign (28 instruments). However, only 12 instruments (from 11 groups) participated in the far lamp measurements. This was made clearer in the abstract and the text. Further, we removed the number of instruments from the abstract (P.1 L.5) and revised section 2.2. As already mentioned above Tables 1 and 2 were merged and the instrument ID was removed.

*P3, L31: There is no need to cite the URL when Kreher et al is already cited.*

The URL was removed from the paper.

*P4, L2: This sentence is redundant.*

This sentence is meant to stress the variety of the different instruments and is kept in the revised version of the paper but in a modified form since Tables 1 and 2 were merged.

*P4, L5: Five Pandoras participated in CINDI-2. Each of them performed sun scans as part of routine operation that served as azimuth and elevation calibration. This method should be also presented for comparison.*

Many thanks for this hint, to which we fully agree. See also answer to major comment 6.

*P4, L16: What is "horizontal line of the telescope"? Is it the optical axis of the telescope/ fiber setup when the instrument points at zero-degree elevation angle? How do you determine it?*

The "horizontal line of the telescope" is defined as the line of sight of the instrument at 0° elevation. This is the property which is actually calibrated by all the methods. This information was added to the paper.

*P4, L30: This is another reason why sun scanning by Pandora instruments should be discussed in this paper.*

See answer to major comment 6.

*P5, L6: How were the distances measured from the lamp to the instruments? How were the vertical distances measured? The land and the canal banks were covered with grass and are not perfectly flat. What is the uncertainty in all distance measurements?*

The horizontal distance was measured using Google Maps, where the location of both the lamp and the measurement site can be clearly identified (see Figure 3). Further, it should be mentioned that the accuracy of the horizontal distance is not critical, since the distance is quite large (more than 1 km). Regarding the vertical distances: they were measured manually using a laser level which was projected onto a folding rule and the channels located next to the lamp and the measurement site. In that way, first the height difference between lamp and the channel's water surface could be determined. Since all channels were connected to each other (except one step which was determined in the same way), the lamp position could be marked on the containers as indicated in figure 3 and described in section 3.2.1. Of course the banks and the land are not perfectly flat, however, the error introduced by that is very small and the overall uncertainty is dominated by measurement errors. The total error in the determination of the lamp mark is estimated to be around 0.2 m which translates to an uncertainty of 0.01° in the lamp position. Additionally, the height differences ($\Delta h$) between the lamp mark and the telescope units of the instruments have to be determined. For this difference an error of around 30 cm is estimated. Therefore the total uncertainty of the lamp position relative to the telescopes is estimated to be 0.5 m which translates to an uncertainty of 0.02° (see also answer to major comment 2). This information as well as revised description of the estimation of lamp positon was added to revised version of the paper.

*P5, L8: Information from Fig 2 can be communicated in Fig 3 and Fig 2 removed.*

Thanks for this idea, however, we do not really agree here. Figure 2 explains the general idea of the elevation calibration procedure which is the same for all methods. Figure 3 sketches the specific application of this general idea to the specific case of the far lamp measurements and gives more details on the setup which was actually used during the campaign. Therefore, we

think figure 3 would be overloaded if we would add the information given in figure 2, and thus we decided to keep both figures in the paper.

*P5, L7: The word "compare" next to most Fig and Table references is unnecessary.*

Thanks for this hint. For the revised version of the paper we checked for those phrases and removed redundant ones.

*P5, L8: The lamp light was collimated and then "directed". How exactly was this achieved? What was the accuracy of the lamp pointing? How uniform was the resulting beam that was visible on the container?*

As described in the paper the lamp was directed towards the instruments using a large aperture lens. Here, the lamp was put in the focal point of the lens which was achieved by minimising the size of the beam (this was already done prior to the campaign). Then the lamp was manually directed towards the campaign site by eye. Here, it should be noted that the exact pointing is not critical as long as the instruments are located within the light cone. We assumed that the diameter of the lens is homogenously bright. Nevertheless, also this assumption is not a critical point, because the angle under which the full lens is seen from the campaign site is <0.01°. We added this description to the revised version of this section.

*P5, L19: Which earlier night, 8 Sep 2016, or before that? Is 0.16 deg the offset from the initial "a prior" calibration done in the lab or from the earlier night?*

Thanks for pointing this like of clarity out. The pre-calibration was done using a water level during the setup of the instrument. Then the finer adjustment was done using the results of the far lamp scans from 7$^{th}$ (in this night the lamp measurements were tested by our group with an scan resolution of 0.1° but the scanning was done manually), 8$^{th}$ and 10$^{th}$ September. The other two nights are then somehow tests of reproducibility. All values in this paper are given relative to the elevation calibration which was obtained by these finer adjustments and which was finally used for the campaign. This information was added to the text.

*P5, L21: Fig 4 is unnecessary and should be removed.*

The exact spectrum of the lamp depends on several properties, e.g. pressure in the lamp and optical filters. Thus, we think it is quite interesting and useful to show a real measured spectrum of the lamp. However, we combined figures 4 and 5 to reduce the numbers of figures in the paper.

*P6, L14: How did you decide that one direction was better than the other? What was positioning error of the lamp? Please include characteristics of instrument positioners (manufacturer accuracy and precision, and used resolution) in Table 2.*

Since also the elevation sequences defined in the synchronised measurement protocol had ascending elevations, we decided on an ascending scanning scheme for all other calibration exercises in order to be consistent. Regarding the positioning error of the lamp, see the answers to major comments 2 and 4. As pointed out above (answer to major comment 7) the positioner accuracy is not the limiting factor of this study, which is dominated by uncertainties related with the determination of the target positions.

*P6, L18: Is 0.16deg the initial calibration? Or is this the effect of the positioner resolution?*

Regarding part 1 of the question, see answer to comment "P5, L19". Regarding part 2: Here we conclude that the 0.16° is an effect of the scan resolution. The positioner resolution is 0.01° for the MPIC instrument.

*P7, L9-10: What is the leveling accuracy of the laser level? What light source is used?*
*How is uniformity of the beam achieved? How accurate is determination of the light source center? What are the requirements of light source installation? It is also assumed that the optical axis of the telescope/fiber setup co-align with the mechanical center of the telescope (e.g. the fiber however can be slightly higher or lower than this estimation). What is the final error in determining this betta (= zero) offset angle between the center of the telescope/fiber and the lamp?*

The accuracy of the laser level has been tested in the lab and amounts to approximately 0.1°. A Hg-Lamp was used. Using a cylinder lens, the laser beam leaves the laser level as a horizontal stripe with a thickness of approximately 2 mm, which is the limiting factor in the determination of the light beam centre (which leads to about 0.04° additional uncertainty in the determination of beta). The stripe has a certain curvature (smile), which is accounted for by only considering the centre of the stripe. This commercial laser level comes with a tripod with adjustable height that can be placed onto any suitable surface. The laser beam creates parallel light that hits the entrance optics approximately at its centre, as controlled by eye. Any vertical displacement of the laser beam is not of importance, since the incoming parallel light is projected onto the focal point of the entrance lens where the fibre entrance is located, independent of any displacement of the beam from the optical axis. The information on the accuracy of the laser level and the error introduced by the beam thickness was added to the paper.

*P7, L18-19: Were all the scans done from the same direction (upwards or downwards).*
*Looking at the intensities for scan 2 and 3 they might be an indicator of the positioner backlash or pointing issues.*

The elevation pointing is continuously regulated by comparison of orientation of the telescope measured by the built-in tilt sensor with the nominal angle. Therefore, there are no backlash effects to be expected. This information was added to the paper.

*P9, L5: It is not quite clear how this FOV determination eliminates dependencies on the scattering conditions (wavelength), underlying surfaces and their albedo, as well as solar position.*

This method determines the "effective" FOV for that specific measurement. Many measurements under different sky conditions give good statistics. The mentioned effects are not critical as long as the horizon is clearly visible and only measurements with favourable conditions (high visibility, no variable and/or low-lying clouds) are used. As described above (answer to major comment 8), this information is provided several times in the manuscript. However, it is stressed more in the revised version of the paper.

*P10, L23. Should "horizon" be replaced with "horizontal"?*

Thanks for this hint. "Horizon" is indeed not correct. Actually, "target positions" fits better. The text was adjusted accordingly.

*P10, L26: How was "visible horizon" determined? How was "closeness" to the visible horizon determined? Figure 3 suggests that the lamp was at 3.3 m above ground. Is "ground level" referring to 3.3 m above ground?*

As "visible horizon" we defined the transition of tree tops to the open sky. We replaced the sentence "…, the xenon lamp was placed close to the visible horizon but at ground level..." by "…, the xenon lamp was placed directly in front of a row of trees which mark the visible horizon (the transition of the tree tops to the open sky)…". Further, the following statement was added to the paper: "The vertical position of the lamp was 3.5 m above the water level in the water channel which was located next to the measurement site (see Fig. 3), and there a few meters below the tree tops." Additionally, the whole paragraph was slightly modified (see also answer to the next comment).

*P10. L29. How was 6.5 to 8.0 m estimated?*

These heights were estimated from the differences between lamp scan and horizon scan results using simple geometry. 0.3° and 0.37° lead to heights of 6.7 m and 8.27 m (at lamp distance), respectively. Since the lamp is in average seen at 0.01° (corresponding to 0.22 m at lamp distance) for the MPIC instrument this leads to heights of 6.5 m and 8.0 m, respectively. These heights are consistent with the (estimated by eye) heights of trees. This is a little bit hand waving but overall consistent and the differences are understandable. A summary of this information was added to the paper.

*P11, L13: I recommend combining Fig 12-15 into one figure. These figures give a good sense of the apparent FOV, as the lamp is scanned, impacted by both optics and precision of the positioner. Selecting the azimuth with the maximum intensity is somewhat arbitrary for some of the instruments with asymmetric FOV (e.g. Fig 13).*

Thanks for this good suggestion. Figures 12-15 were combined. The reviewer is right that the selection of the azimuth with the maximum intensity is somehow arbitrary. However, this choice is not critical for the interpretation of the results.

*P11, L20: Differences in positioner pointing precision is also an important parameter of the apparent FOV.*

Thanks for this good comment. This information was added to the text.

*P11, L21: Fig 12 suggests that FWHM is about 0.65 deg for the BIRA instrument not 1 deg.*

Thanks for this hint. As mentioned at the first minor comment, we added a section on the comparison between the retrieved FOVs (from horizon and lamp scans) and the reference FOV. There, smaller retrieved FOVs were found compared to the ones provided by the groups. However, as explained in the answer of the next comment the mentioned passage was removed from the paper.

*P11, L22: I would say good alignment, center "spot" fiber arrangement and good positioning are the reasons for a relatively uniform FOV. Unless the lamb beam was not uniform.*

Thanks for this good comment. We already mentioned the fibre arrangement in the paper, however, did not stress this enough. The text was adjusted accordingly. Further, the quality of the positioning was added to the text. After discussion with the co-authors, the influence of the size of the FOV on the smoothness of the intensity distribution was removed.

*P12, L23: Isn't the spread expected due to differences in prior reference calibration of 0 elevation angle?*

Thanks for the comment. We think it refers to P.13 L.3. The reviewer is right, the spread is expected due to differences in the prior reference calibration of the 0° angle as explained in the text (P.13 L.12-17).

*P13, L9: Precision (repeatability) will be more appropriate here instead of "accuracy".*

Thanks for this hint. The text was adjusted accordingly.

*P13, L16: I would replace "measure of consistency" with: "measure of variability".*

We agree, the text was adjusted accordingly.

*P13, L22-23: How were 6.5 and 8 m as a function of wavelength (340 and 440 nm) estimated?*

They were estimated using the horizon elevations of the MPIC instrument as described on P.10 L.29-32. A more detailed description can be found in the answer to the comment P10. L29 and was added to the text in section 3.6 (3.7 in the revised version). Further, a cross reference to this section was added in the revised version of the paper.

*P14, L32: Reading the text that follows you assert this difference is due to surface reflectivity. I also will add effects of FOV and wavelength dependent scattering. For a system with a 0.6 deg field of view placed directly on the ground pointing at 0 deg elevation angle (assuming no obstacles): half of the FOV will receive photons scattered in the atmosphere and half reflected from the underlying surfaces. Since Rayleigh scattering is wavelength dependent more photons at longer wavelength would be scattered in the "above ground" half of FOV. So for the instruments with FOV ~0.6 nm FWHM the telescopes should point at least 0.3 deg below horizon to minimize the effect of FOV size. Considering that the instruments during CINDI-2 were located ~4 m and 7 m above ground this angle should be even larger and depends on the wavelength and distance to obstacles and solar position (SZA and RAA).*

Thanks for this comment and the detailed explanation. However, radiative transfer simulations confirmed that the wavelength dependence of scattering cannot explain the observed wavelength dependence of the elevations scans. This information was added to the paper and the whole part was shortened.

*P14, L29: Figure 19 does not support this statement. BIRA_4 instrument horizon position is about the same for both wavelengths.*

Many thanks for this hint. We corrected the text accordingly.

*P15, L25-31: It is unclear how apparent horizon measurements can be suitable for pointing accuracy calibration.*

Indeed the horizon scans have to be considered with care. But as long as the apparent position of the horizon is known, horizon scans are a valuable tool for elevation calibration. However, only measurements under favourable conditions (high visibility, no rapidly varying and/or low-lying clouds) should be used. As mentioned in the answer to major comment 8, we added these

precaution measures which should be fulfilled to the text (horizon scans and conclusions sections). Further, in this text passage "apparent" was changed to "visible".

*P16, L5: The conclusion maybe applies to the better performing instruments, while the rest of them mostly excluded from the analysis.*

This conclusion is derived from the regression analyses which included all instruments which performed lamp measurements and horizon scans. Naturally, some perform better than others. Reasons for deviations were assessed in sections 4.1, 4.2, 4.3, however, these descriptions were slightly modified for the revised version of the paper. Nevertheless, the results of most of the instruments (for which both methods were applied) are rather close to the fitted line.

*P16, L12-13: I do not agree that all the instruments showed consistent results. TLS mainly derived dependence for the better performing instruments... Also the authors have not demonstrated that the instruments improved their pointing performance as a result of any of these calibration methods.*

The reviewer is right, the TLS fit weights the different data points with respect to their individual errors. However, Figures 24 and 25 show that the results from many of the instruments scatter around the regression line, except 1-2 worse performing instruments (for which the lower weighting in the regression is justified). In summary, given the uncertainties (which are discussed in the paper) and the different instrument performances, we think that this statement is still justified. We replaced the word "all" by the word "most". Regarding the second point: This is true. However, as mentioned above, the focus of this paper was on the details of the elevation calibration. The assessment of the performance during the measurement campaign is described in Kreher et al. (2019).

*P16, L33: I would not call this method accurate since some of the instruments clearly showed asymmetric FOV and different functions other than Gaussian could describe the intensity distribution potentially leading to larger errors.*

Fit errors were added to all plots showing Gaussian fits. They indicate that the standard fitting errors are rather small despite the fact that some intensity distributions do not strictly follow a Gaussian shaped curve. As one conclusion of this study it was found that the centre of the Gaussian is a rather robust measure which might be different for the other parameters which are fitted (e.g. amplitude of the Gaussian). This becomes clearer when comparing the fitted centres with the centres retrieved from a centre of mass approach (see section 3.2.2). Nevertheless, the sentence was changed to "Furthermore, this method is very accurate and precise as long as the instrument has a mostly symmetric FOV".

*P17, L1: Precision should not be confused with accuracy. In my opinion the authors have not accounted for all the uncertainties to claim accuracy of +-0.05 deg.*

As mentioned previously (answers to major comments 3 and 7) the systematic errors are dominated by the errors of determination of the lamp position and the fit error. However, for the setup and the measurements used in this study this estimate is justified. But the reviewer is right that for different locations the errors might be larger and therefore, the method might be less accurate.

*I recommend combining Table 1 and 2 and adding 2 columns with positioner maker, accuracy and precision data.*

The tables were merged in the revised version of the paper.

*I do not think Tables 3 and 4 are needed.*

We prefer to keep these two tables in the paper to provide a clear summary of the results. Note that these tables are small and don't consume much space.

*Table 5: I think replacing "row" with "container level" might be clearer.*

We agree and adjusted the table accordingly.

*Figure 2 and 3 should be combined.*

We prefer to keep these two figures separately as explained above.

*Figure 3: Upper and lower "panels" instead of "parts".*

The caption and the text were adjusted accordingly.

*Figure 4 is unnecessary*

We prefer to keep this figure as explained above.

*Figure 10: It might be good to raise MAX-DOAS instrument in line with the stripe*

Thanks for this good suggestion. The figure was modified accordingly.

*Figures 12, 13, 14, 15 should be combined to give better idea about apparent FOV for different instruments.*

We agree and combined the figures.

*Figure 16: intensity distributions in (a, b) clearly do not have Gaussian distributions, please add fitting errors.*

Standard fitting errors were added to all plots showing Gaussian fits.

*Figures 24 and 25: Y-axis = Apparent horizon elevation; X-axis = apparent lamp elevation. Add fitting errors to slope and intercept.*

The two figures were modified accordingly. Further, the standard error of the mean lamp and horizon elevations are now shown instead of the standard deviations.

---

## Author Response (AR2)

Dear Editor,

We are glad to submit a revised version of the manuscript "Evaluating different methods for elevation calibration of MAX-DOAS instruments during the CINDI-2 campaign" (amt-2019-115).

The raised comments have been accounted for in the revised version of the manuscript. Further, the text has been checked for unnecessary connective words and some of them have been removed. Regarding possible additional grammar mistakes we refer to corrections during typesetting.

We provide an updated version of the manuscript, as well as a version with tracked-changes in order to expose the modifications we made.

Below we have attached

1, the replies to the editor comments, and

2, a revised version of the manuscript with tracked-changes.

Kind regards,

Sebastian Donner

**Reply to comments from the Associate Editor**

We would like to thank the editor for this positive assessment and several helpful suggestions to further improve the manuscript.

**Minor Comments:**

*1) Page 4, line 28: Please remove comma after "Others, ".*

Done.

*2) Page 5, line 8: The newly added text "The horizontal line corresponds to the ….." is out of context here. Please move this statement to the figure caption of Fig.2.*

Done.

*3) Page 5, line 18: Add comma between "Others" and "however"*

"However" was removed from the sentence.

*4) Page 12, line 20: Please remove "even" from "The PANDORA systems even …." as tracking sun is the characteristic of the instrument.*

Done.

*5) Page 14, line 13: Please remove "also" in "Further, also differences …".*

Done.

*6) Page 16, line 13: Please replace "Like explained" by "As explained".*

Done. Further, the sentence "As mentioned above, during the day the visible horizon might be defined by obstacles such as trees." was changed to "Already above it was mentioned that during the day the visible horizon might be defined by obstacles such as trees.", to avoid the same words at the beginning of the two sentences.

*7) Page 20, line 17: The statement "Finally, the sun scans were described" is unnecessary; please, delete it.*

Done.

*8) Page 20, line 31: You may want to change " the (systematical) error" to "the systematic error".*

Done.

*9) Page 21, lines 30-31: Revise this statement "Consistent results were found … and also here…". It may be a good idea to split the statement into two sentences.*

The statement was revised and split into two sentences.

*10) Page 22, line 12: The sentence (the new text) is incomplete. Please revise.*

The sentence was revised.

*11) Page 22, line 28: Please replace "Finally, sun scans were described and explained. Here" by " As discussed in Sect. 3.6"*

Done.

*12) Page 22, lines 34-35. The last part ("and the rather high complexity") of this statement is unclear. Please, revise the statement.*

The sentence was revised. Since the higher complexity of this method is already apparent by the description of this method and to avoid confusion we decided to remove this statement from the paper and from Table 5.

*13) Page 32, Table 5 caption: Please replace "Columns 4 and 5 resume" by "Columns 4 and 5 list".*

Done.

*14) Page 36, Figure 4: The text says "The obtained center of mass is indicated by the blue dashed line", but I have problem to locate the blue dashed line in the figure. Please, check. Also, "For the explanation of the specific shape …." is unnecessary as including figure is the sole idea.*

The editor is right that it is quite hard to locate the blue dashed line since it is basically located at the same x-value (elevation angle of 0.02 °) as the red dashed line (center of Gaussian fit). A close zoom shows that. Other colours were tested which did not really solve this issue. Because of that and in order to have the same colours for all plots we decided to keep the figure unchanged. The statement "For the explanation of the specific shape of the intensity curve, see text." Was removed from the caption.

*15) Page 47, Figure 15 caption: Please replace "25 %" by "25th" and "75 %" by "75th" as you spell percentiles after those numbers.*

Done.

*16) Page 54, Table A1: Typo for "and" in 4th row and 2nd column. Suggest cut "(good an worse ones)". Revise the statement for the 2nd part of 6th row and 3rd column "If only good .." for clarity.*

Done.

*17) Page 55, Table A2: Typo for "than" in 4th row and 3rd column ("longer that" should be "longer than"). Please cut "out" in its last sentence as it is unnecessary.*

Done.

[revised manuscript text omitted]